# Lactate is an energy substrate for rodent cortical neurons and enhances their firing activity

Anastassios Karagiannis[1], Thierry Gallopin[2], Alexandre Lacroix[1], Fabrice Plaisier[1], Juliette Piquet[1], Hélène Geoffroy[2], Régine Hepp[1], Jérémie Naudé[1], Benjamin Le Gac[1], Richard Egger[3], Bertrand Lambolez[1], Dongdong Li[1], Jean Rossier[1,2], Jochen F Staiger[4], Hiromi Imamura[5], Susumu Seino[6], Jochen Roeper[3], Bruno Cauli[1]*

[1]Sorbonne Université, CNRS, INSERM, Neurosciences Paris Seine - Institut de Biologie Paris Seine (NPS-IBPS), Paris, France; [2]Brain Plasticity Unit, CNRS UMR 8249, CNRS, ESPCI Paris, Paris, France; [3]Institute for Neurophysiology, Goethe University Frankfurt, Frankfurt, Germany; [4]Institute for Neuroanatomy, University Medical Center Göttingen, Georg-August- University Göttingen, Goettingen, Germany; [5]Graduate School of Biostudies, Kyoto University, Kyoto, Japan; [6]Division of Molecular and Metabolic Medicine, Kobe University Graduate School of Medicine, Hyogo, Japan

*For correspondence:
bruno.cauli@upmc.fr

Competing interest: The authors declare that no competing interests exist.

**Abstract** Glucose is the mandatory fuel for the brain, yet the relative contribution of glucose and lactate for neuronal energy metabolism is unclear. We found that increased lactate, but not glucose concentration, enhances the spiking activity of neurons of the cerebral cortex. Enhanced spiking was dependent on ATP-sensitive potassium ($K_{ATP}$) channels formed with KCNJ11 and ABCC8 subunits, which we show are functionally expressed in most neocortical neuronal types. We also demonstrate the ability of cortical neurons to take-up and metabolize lactate. We further reveal that ATP is produced by cortical neurons largely via oxidative phosphorylation and only modestly by glycolysis. Our data demonstrate that in active neurons, lactate is preferred to glucose as an energy substrate, and that lactate metabolism shapes neuronal activity in the neocortex through $K_{ATP}$ channels. Our results highlight the importance of metabolic crosstalk between neurons and astrocytes for brain function.

## Editor's evaluation

This manuscript shows that lactate is an important energy substrate and its metabolism shapes neuronal activity in the rodent somatosensory neocortex through $K_{ATP}$ channels.

## Introduction

The human brain represents 2% of the body mass, yet it consumes about 20% of blood oxygen and glucose which are mandatory energy substrates (*Clarke and Sokoloff, 1999*). The majority (~50–80%) of the cerebral energy metabolism is believed to be consumed by the Na$^+$/K$^+$ ATPase pump to maintain cellular ionic gradients dissipated during synaptic transmission and action potentials (*Attwell and Laughlin, 2001*; *Lennie, 2003*). Synaptic and spiking activities are also coupled with local cerebral blood flow and glucose uptake (*Devor et al., 2008*; *Logothetis, 2008*). This process, referred to as neurovascular and neurometabolic coupling, is the physiological basis of brain imaging

techniques (*Raichle and Mintun, 2006*) and maintains extracellular glucose within a physiological range of 2–3 mM (*Silver and Erecińska, 1994*; *Hu and Wilson, 2002*). Also, following increased neuronal activity extracellular lactate increases (*Prichard et al., 1991*; *Hu and Wilson, 1997*) for several minutes up to twice of its 2–5 mM basal concentration despite oxygen availability (*Magistretti and Allaman, 2018*).

Based on the observations that various byproducts released during glutamatergic transmission stimulate astrocyte glucose uptake, aerobic glycolysis, and lactate release (*Pellerin and Magistretti, 1994Voutsinos-Porche et al., 2003* ; *Ruminot et al., 2011*; *Choi et al., 2012*; *Sotelo-Hitschfeld et al., 2015*; *Lerchundi et al., 2015*), lactate has been proposed to be shuttled from astrocytes to neurons to meet neuronal energy needs. This hypothesis is supported by the existence of a lactate gradient between astrocytes and neurons (*Mächler et al., 2016*), the preferential use of lactate as an energy substrate in cultured neurons (*Bouzier-Sore et al., 2003*; *Bouzier-Sore et al., 2006*), and its ability to support neuronal activity during glucose shortage (*Schurr et al., 1988*; *Rouach et al., 2008*; *Wyss et al., 2011*; *Choi et al., 2012*). However, the use of different fluorescent glucose analogs to determine whether astrocytes or neurons take up more glucose during sensory-evoked neuronal activity has led to contradicting results (*Chuquet et al., 2010*; *Lundgaard et al., 2015*). Furthermore, brain slices and in vivo evidence have indicated that synaptic and sensory stimulation enhanced neuronal glycolysis and potentially lactate release by neurons (*Ivanov et al., 2014*; *Díaz-García et al., 2017*), thereby challenging the astrocyte–neuron lactate shuttle hypothesis. Hence, the relative contribution of glucose and lactate to neuronal ATP synthesis remains unresolved.

ATP-sensitive potassium channels ($K_{ATP}$) act as metabolic sensors controlling various cellular functions (*Babenko et al., 1998*). Their open probability ($p_o$) is regulated by the energy charge of the cell (i.e., the ATP/ADP ratio). While ATP mediates a tonic background inhibition of $K_{ATP}$ channels, cytosolic increases of ADP concentrations that occur as a sequel to enhanced energy demands, increase the $p_o$ of $K_{ATP}$ channels. In neurons, electrical activity is accompanied by enhanced sodium influx, which in turn activates the $Na^+/K^+$ ATPase. Activity of this pump alters the submembrane ATP/ADP ratio sufficiently to activate $K_{ATP}$ channels (*Tanner et al., 2011*). The use of fluorescent ATP/ADP biosensors has demonstrated that $K_{ATP}$ channels are activated ($p_o > 0.1$) when ATP/ADP ratio is ≤5 (*Tantama et al., 2013*).

$K_{ATP}$ channels are heterooctamers composed of four inwardly rectifying $K^+$ channel subunits, KCNJ8 (previously known as Kir6.1) or KCNJ11 (previously known as Kir6.2), and four sulfonylurea receptors, ABCC8 (previously known as SUR1) or ABCC9 (previously known as SUR2), the later existing in two splice variants (SUR2A and SUR2B) (*Sakura et al., 1995*; *Aguilar-Bryan et al., 1995*; *Aguilar-Bryan et al., 1995* ; *Inagaki et al., 1995b* ; *Isomoto et al., 1996*; *Inagaki et al., 1996* ; *Chutkow et al., 1996*; *Yamada et al., 1997*; *Li et al., 2017*; *Martin et al., 2017*; *Chen, 2017*; *Puljung, 2018*). The composition in $K_{ATP}$ channel subunits confers different functional properties, pharmacological profiles as well as metabolic sensitivities (*Isomoto et al., 1996Inagaki et al., 1996*; *Gribble et al., 1997*; *Yamada et al., 1997*; *Okuyama et al., 1998*; *Liss et al., 1999*). $K_{ATP}$ channel subunits are expressed in the neocortex (*Ashford et al., 1988*; *Karschin et al., 1997*; *Dunn-Meynell et al., 1998*; *Thomzig et al., 2005*; *Cahoy et al., 2008*; *Zeisel et al., 2015*; *Tasic et al., 2016*) and have been shown to protect cortical neurons from ischemic injury (*Héron-Milhavet et al., 2004*; *Sun et al., 2006*) and to modulate their excitability (*Giménez-Cassina et al., 2012*) and intrinsic firing activity (*Lemak et al., 2014*). $K_{ATP}$ channels could thus be leveraged to decipher electrophysiologically the relative contribution of glucose and lactate to neuronal ATP synthesis. Here, we apply single-cell RT-PCR (scRT-PCR) to identify the mRNA subunit composition of $K_{ATP}$ channel across different neocortical neuron subtypes and demonstrate lactate as the preferred energy substrate that also enhances firing activity.

## Results

### Expression of $K_{ATP}$ channel subunits in identified cortical neurons

We first sought to determine whether $K_{ATP}$ channel subunits were expressed in different neuronal subtypes from the neocortex. Neurons ($n$ = 277) of the juvenile rat barrel cortex from layers I to IV (*Supplementary file 1*) were functionally and molecularly characterized in acute slices by scRT-PCR (*Figure 1*), whose sensitivity was validated from 500 pg of total cortical RNAs (*Figure 1—figure supplement 1A*). Neurons were segregated into seven different subtypes according to their overall

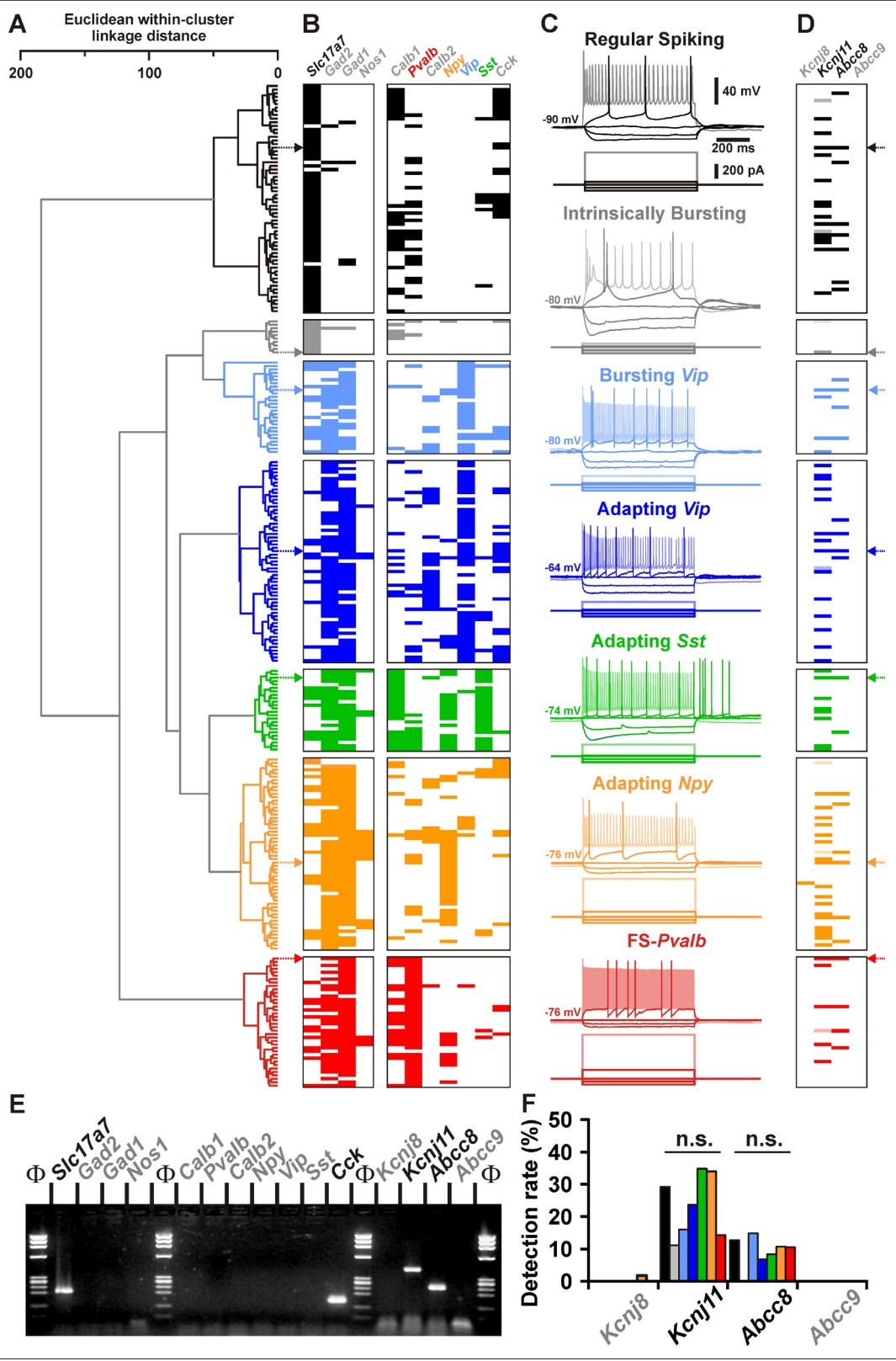

**Figure 1.** Detection of *Kcnj11* and *Abcc8* K<sub>ATP</sub> channel subunits in cortical neuron subtypes. (**A**) Ward's clustering of 277 cortical neurons (left panel). The x-axis represents the average within-cluster linkage distance, and the y-axis the individuals. (**B**) Gene detection profile across the different cell clusters. For each cell, colored and white rectangles indicate presence and absence of genes, respectively. (**C**) Representative voltage responses induced by

*Figure 1 continued on next page*

*Figure 1 continued*

injection of current pulses (bottom traces) corresponding to −100, −50, and 0 pA, rheobase and intensity inducing a saturating firing frequency (shaded traces) of a regular spiking neuron (black), an intrinsically bursting neuron (gray), a bursting vasoactive intestinal polypeptide (*Vip*) interneuron (light blue), an adapting *Vip* interneuron (blue), an adapting *Sst* interneuron (green), an adapting *Npy* interneuron (orange), and a **Fast Spiking-Parvalbumin interneuron** (FS-*Pvalb*, red). The colored arrows indicate the expression profiles of neurons whose firing pattern is illustrated in (**C**). (**D**) Detection of the subunits of the $K_{ATP}$ channels in the different clusters. Shaded rectangles represent potential *Kcnj11* false positives in which genomic DNA was detected in the harvested material. (**E**) Single-cell RT-PCR (scRT-PCR) analysis of the regular spiking (RS) neuron depicted in (**A–D**). (**F**) Histograms summarizing the detection rate of $K_{ATP}$ channel subunits in identified neuronal types. n.s., not statistically significant.

The online version of this article includes the following figure supplement(s) for figure 1:

**Source data 1.** Somatic, electrophysiological, and molecular properties of the cortical neurons shown in *Figure 1A–D*.

**Source data 2.** Original file of the full raw unedited gel shown in *Figure 1E*.

**Source data 3.** Uncropped gel shown in *Figure 1E* with relevant bands labeled.

**Source data 4.** Statistcal comparisons of the detection of $K_{ATP}$ channel subunits in different types of cortical neurons shown in *Figure 1F*.

**Figure supplement 1.** Molecular expression of $K_{ATP}$ channels.

**Figure supplement 1—source data 1.** Original file of the full raw unedited gel shown in *Figure 1—figure supplement 1A*.

**Figure supplement 1—source data 2.** Uncropped gel shown in *Figure 1—figure supplement 1A* with relevant lanes labeled.

**Figure supplement 1—source data 3.** Original file of the full raw unedited gel shown in *Figure 1—figure supplement 1B*.

**Figure supplement 1—source data 4.** Uncropped gel shown in *Figure 1—figure supplement 1B* with relevant bands labeled.

molecular and electrophysiological similarity (*Figure 1A*) using unsupervised Ward's clustering (*Ward, 1963*), an approach we previously successfully used to classify cortical neurons (*Cauli et al., 2000*; *Gallopin et al., 2006*; *Karagiannis et al., 2009*). Regular spiking (RS, *n* = 63) and intrinsically bursting (IB, *n* = 10) cells exhibited the molecular characteristics of glutamatergic neurons, with very high single-cell detection rate (*n* = 69 of 73, 95%) of vesicular glutamate transporter 1 (*Slc17a7*) and low detection rate (*n* = 7 of 73, 10%) of glutamic acid decarboxylases (*Gad*s, *Figure 1B–E* and *Supplementary file 2*), the GABA synthesizing enzymes. This group of glutamatergic neurons distinctly displayed hyperpolarized resting membrane potential (−81.2 ± 0.8 mV), possessed a large membrane capacitance (108.6 ± 3.6 pF), discharged with wide action potentials (1.4 ± 0.0 ms) followed by medium afterhyperpolarizations (mAHs). These neurons did sustain only low maximal frequencies (35.4 ± 1.6 Hz) and showed complex spike amplitude accommodation (*Supplementary file 5*). In contrast to RS neurons, IB neurons were more prominent in deeper layers (*Supplementary file 1*) and their bursting activity affected their adaptation amplitudes and kinetics (*Figure 1C* and *Supplementary files 4 and 5*), spike broadening (*Figure 1C* and *Supplementary file 6*), and the shape of mAHs (*Figure 1C* and *Supplementary file 7*).

All other neuronal subtypes were characterized by a high single-cell detection rate of *Gad2* and/or *Gad1* mRNA (*n* = 202 of 204, 99%, *Figure 1B* and *Supplementary file 2*) and therefore likely corresponded to GABAergic interneurons. Among *Gad*-positive population, neurons were frequently positive for vasoactive intestinal polypeptide (*Vip*) mRNA, and in accordance to their electrophysiological phenotypes, were segregated into bursting *Vip* (*n* = 27) and adapting *Vip* (*n* = 59) neurons. These *Vip* interneurons were further characterized by high membrane resistance (581 ± 27 MΩ) and small membrane capacitance (52.7 ± 2.3 pF, *Figure 1B, C* and *Supplementary file 3*).

In other GABAergic interneurons somatostatin (*Sst*) and calbindin (*Calb1*) as well as neuropeptide Y (*Npy*) to a lesser extent were frequently detected and functionally corresponded to Adapting *Sst* neurons (*n* = 24, *Figure 1B* and *Supplementary file 2*). They displayed depolarized resting membrane potential, pronounced voltage sags, low rheobases, and pronounced afterdepolarizations

(ADs; *Figure 1C* and *Supplementary files 3; 4 and 7*). In another group of GABAergic adapting interneurons located in superficial layers, mRNA for *Npy* was detected at a high rate (*n* = 31 of 56, 55%). In these Adapting NPY interneurons mRNA for nitric oxide synthase-1 (*Nos1*) was detected at a lower rate (*Figure 1B* and *Supplementary files 1 and 2*). In response to suprathreshold depolarizing current steps, these interneurons showed very little spike frequency adaptation (*Figure 1C* and *Supplementary file 4*). Finally, parvalbumin (*Pvalb*) was observed in virtually all neurons of a subpopulation termed Fast Spiking-*Pvalb* interneurons (FS-*Pvalb*, *n* = 37 of 38, 97%, *Figure 1B* and *Supplementary file 2*). In comparison to all other cortical neurons described above, they were characterized by low membrane resistance (201 ± 13 MΩ), fast time constant, high rheobase, very short spikes (0.6 ± 0.0 ms) with sharp fast afterhyperpolarizations (fAHs) and the ability to sustain high firing rates (139.9 ± 6.8 Hz) with little to no frequency adaptation (*Figure 1C* and *Supplementary files 3-7*). These data thus identified different neuronal subtypes based on their distinctive electrophysiological and molecular features (*Petilla Interneuron Nomenclature Group et al., 2008*) confirming our previous classification schemes (*Cauli et al., 2000*; *Gallopin et al., 2006*; *Karagiannis et al., 2009*).

The functional and molecular classification of cortical neurons allowed us to probe for the single-cell expression of mRNA for $K_{ATP}$ channel subunits (*Figure 1—figure supplement 1A*) in well defined subpopulations. Apart from a single Adapting *Npy* neuron (*Figure 1D*), where *Kcnj8* mRNA was observed, only the *Kcnj11* and *Abcc8* subunits were detected in cortical neurons (in 25%, *n* = 63 of 248 neurons; and in 10%, *n* = 28 of 277 of neurons; respectively). The single-cell detection rate was similar between the different neuronal subtypes (*Figure 1F*). We also codetected *Kcnj11* and *Abcc8* in cortical neurons (*n* = 14 of 248, *Figure 1D*) suggesting the expression of functional $K_{ATP}$ channels.

## Characterization of $K_{ATP}$ channels in cortical neurons

To assess functional expression of $K_{ATP}$ channels in identified cortical neurons (*n* = 18, *Figure 2A*), we measured the effects of different $K_{ATP}$ channel modulators on whole-cell currents ($Q_{(3,18)}$ = 32.665, p = 3.8 × $10^{-7}$, Friedman test) and membrane resistances ($Q_{(3,18)}$ = 40.933, p = 6.8 × $10^{-9}$). Pinacidil (100 µM), an ABCC9-preferring $K_{ATP}$ channel opener (*Inagaki et al., 1996*; *Moreau et al., 2005*), had little or no effect on current (4.1 ± 3.7 pA, p = 0.478) and membrane resistance (−9.6 ± 3.7%, p = 0.121, *Figure 2B, C*). By contrast, diazoxide (300 µM), an opener acting on ABCC8 and SUR2B-containing $K_{ATP}$ channels (*Inagaki et al., 1996*; *Moreau et al., 2005*), consistently induced an outward current (45.0 ± 9.6 pA, p = 4.8 × $10^{-5}$) and a decrease in membrane resistance (−34.5 ± 4.3%, p = 3.6 × $10^{-5}$) indicative of the activation of a hyperpolarizing conductance (*Figure 2B, C*). The sulfonylurea tolbutamide (500 µM, *Figure 2B–C*), a $K_{ATP}$ channel blocker (*Ammälä et al., 1996*; *Isomoto et al., 1996*; *Gribble et al., 1997*; *Isomoto and Kurachi, 1997*), did not change whole-cell basal current (−6.6 ± 3.0 pA, p = 0.156) or membrane resistance (20.5 ± 7.5%, p = 3.89 × $10^{-2}$). Conversely, tolbutamide dramatically reversed diazoxide ffects on both current (p = 4.1 × $10^{-8}$) and membrane resistance (p = 5.8 × $10^{-10}$).

All pharmacologically analyzed neurons (*n* = 63) exhibited a more positive whole-cell current (Δ*I* = 53 ± 6 pA, range: 4–228 pA) and a lower membrane resistance (Δ$R_m$ = −270 ± 31 MΩ, range: −17 to −1221 MΩ) under diazoxide than under tolbutamide, indicative of their sensitivy to $K_{ATP}$ channel manipulation. In virtually all neuronal subtypes ($H_{(6,43)}$ = 2.274, p = 0.810, Kruskal–Wallis *H* test) or groups ($t_{(42)}$ = 0.3395, p = 0.736, Student's *t*-test), the diazoxide–tolbutamide current/voltage relationship reversed very close to the theoretical potassium equilibrium potential ($E_K$ = −106.0 mV, *Figure 2D–F*) confirming the opening of a selective potassium conductance. Besides its effects on plasma membrane $K_{ATP}$ channels, diazoxide is also a mitochondrial uncoupler (*Dröse et al., 2006*) which increases reactive oxygen species (ROS) production. This might stimulate $Ca^{2+}$ sparks and large-conductance $Ca^{2+}$-activated potassium channels (*Xi et al., 2005*) leading to potential confounding effects. This possibility was ruled out by the observation that Mn(III)tetrakis(1-methyl-4-pyridyl)porphyrin (MnTMPyP, 25 µM), a ROS scavenger (*D'Agostino et al., 2007*), did not reduce the diazoxide–tolbutamide responses on current ($t_{(10)}$=0.76559, p = 0.462, *Figure 2—figure supplement 1A, B*) and conductance ($t_{(10)}$=1.24758, p = 0.241, *Figure 2—figure supplement 1C*).

Cortical neurons exhibited $K_{ATP}$ conductances of similar value between their subtypes ($H_{(6,63)}$ = 5.6141, p = 0.468) or groups ($U_{(9,54)}$ = 233, p = 0.855, Mann–Whitney *U* test, *Figure 2—figure supplement 2 A,B*). $K_{ATP}$ channels activated by diazoxide essentially doubled the whole-cell conductance in the subthreshold membrane potential compared to control or tolbutamide conditions, regardless of

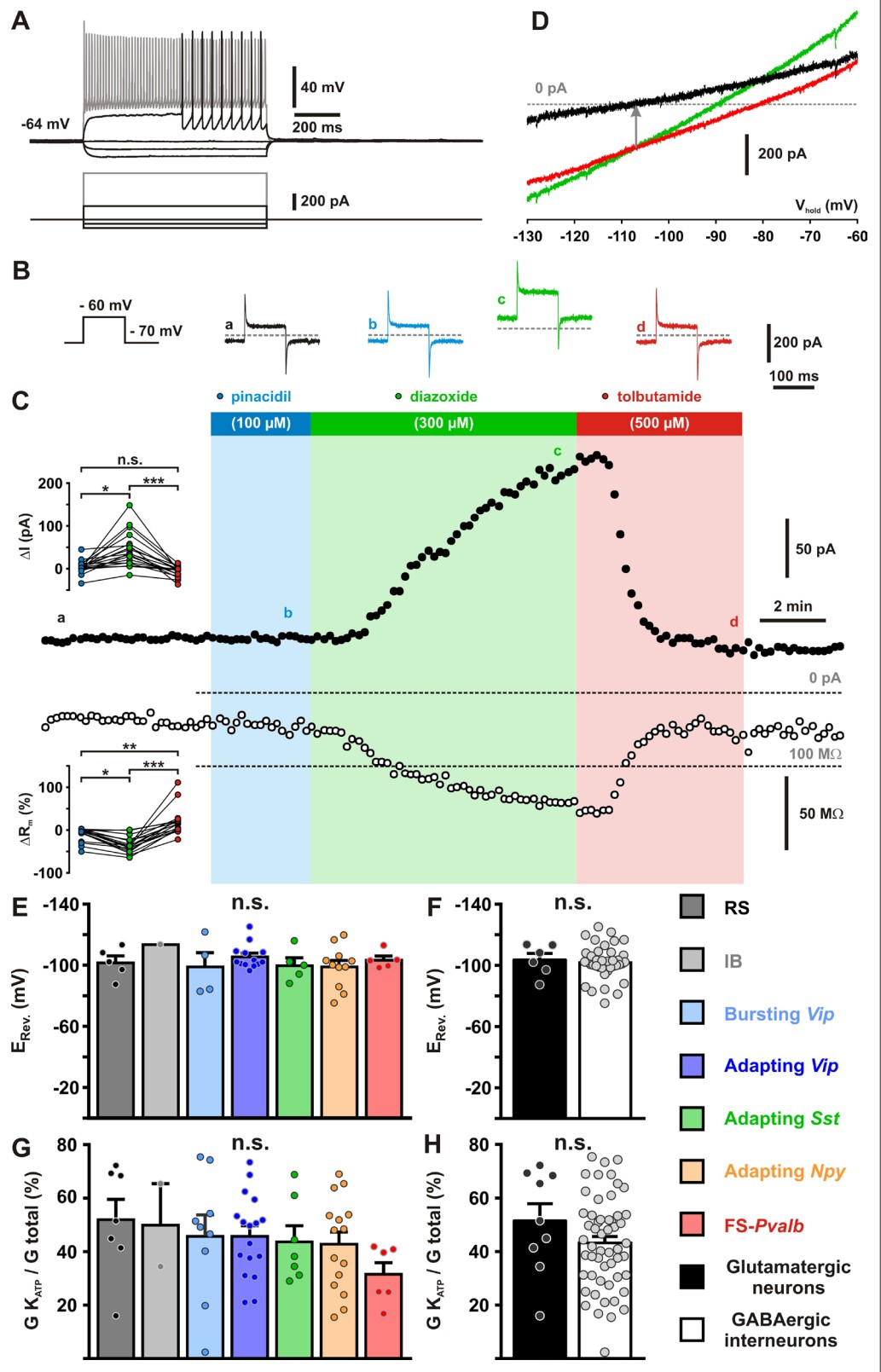

**Figure 2.** Pharmacological and biophysical characterization of K$_{ATP}$ channels in cortical neurons. (**A**) Representative voltage responses of a Fast Spiking-Parvalbumin (FS-*Pvalb*) interneuron induced by injection of current pulses (bottom traces). (**B**) Protocol of voltage pulses from −70 to −60 mV (left trace). Responses of whole-cell currents in the FS-*Pvalb* interneurons shown in (**A**) in control condition (black) and in presence of pinacidil (blue), piazoxide

*Figure 2 continued on next page*

*Figure 2 continued*

(green) and tolbutamide (red) at the time indicated by a–d in (**C**). (**C**) Stationary currents recorded at −60 mV (filled circles) and membrane resistance (open circles) changes induced by $K_{ATP}$ channel modulators. The colored bars and shaded zones indicate the duration of application of $K_{ATP}$ channel modulators. Upper and lower insets: changes in whole-cell currents and relative changes in membrane resistance induced by $K_{ATP}$ channel modulators, respectively. (**D**) Whole-cell current–voltage relationships measured under diazoxide (green trace) and tolbutamide (red trace). $K_{ATP}$ *I/V* curve (black trace) obtained by subtracting the curve under diazoxide by the curve under tolbutamide. The arrow indicates the reversal potential of $K_{ATP}$ currents. Histograms summarizing the $K_{ATP}$ current reversal potential (**E, F**) and relative $K_{ATP}$ conductance (**G,H**) in identified neuronal subtypes (**E, G**) or between glutamatergic and GABAergic neurons (**F, G**). Data are expressed as mean ± standard error of the mean (SEM), and the individual data points are depicted. n.s., not statistically significant. *, ** and *** indicate statistically significant with p< 0.05, 0.01 and 0.001 respectively.

The online version of this article includes the following source data and figure supplement(s) for figure 2:

**Source data 1.** Statistical analyses of whole-cell current and membrane resistance changes induced by $K_{ATP}$ channel modulators (shown in *Figure 2C* insets).

**Source data 2.** Statistical comparisons of $K_{ATP}$ current reversal potential and relative $K_{ATP}$ conductance between neuronal subtypes and groups (shown in *Figure 2E–H*) and of whole-cell $K_{ATP}$ conductance and current density (shown in *Figure 2—figure supplement 2*).

**Figure supplement 1.** Diazoxide-induced current is independent of reactive oxygen species (ROS) production.

**Figure supplement 1—source data 1.** Statistical analyses of the effect of MnTMPyP on normalized $K_{ATP}$ currents and conducatnce whole-cell KATP conductance (shown in *Figure 2—figure supplement 2B,C*).

**Figure supplement 2.** Characterization of $K_{ATP}$ channels in different cortical neurons.

---

neuronal subtypes ($H_{(6,63)}$ = 5.4763, p = 0.484) or groups ($t_{(61)}$ = 1.324, p = 0.191, *Figure 2G, H*). Also, $K_{ATP}$ current density was similar ($H_{(6,63)}$ = 4.4769, p = 0.612, $U_{(9,54)}$ = 240.5, p = 0.965, *Figure 2—figure supplement 2D*). Twenty-nine diazoxide/tolbutamide-responsive neurons were successfully characterized by scRT-PCR. *Kcnj11* and *Abcc8* mRNAs were detected in 35 % (*n* = 10 of 29) and 7 % (*n* = 2 of 29) of these neurons, respectively. These proportions are low compared to the pharmacological responsiveness but similar to the whole sample of profiled cortical neurons (p = 0.3721 and p = 1.0000, Fisher's exact test). These observations suggest that *Kcnj11* and *Abcc8* subunits were underdetected by scRT-PCR mRNA profiling. Together with the pinacidil unresponsiveness and the lack of *Abcc9* detection, these data indicate that the large majority of cortical neurons express functional ABCC8-mediated $K_{ATP}$ channels across different subpopulations. To confirm that KCNJ11 is the poreforming subunit of $K_{ATP}$ channels in cortical neurons, we used a genetic approach based on *Kcnj11* knockout mice (*Miki et al., 1998*). We first verified that *Kcnj11* and *Abcc8* subunits can be detected in pyramidal cells from wild-type mice by scRT-PCR (*Figure 3A, B*). We next used a dialysis approach by recording neurons with an ATP-free pipette solution (*Miki et al., 2001*) enriched in sodium (20 mM) to stimulate submembrane ATP depletion and ADP production by the $Na^+/K^+$ ATPase, which is known to activate $K_{ATP}$ channels (*Figure 3H*). We confirmed that *Atp1a1* and *Atp1a3* were the main α-subunits of the $Na^+/K^+$ ATPase pump detected in pyramidal neurons (*Zeisel et al., 2015*; *Tasic et al., 2016*). Dialysis of ATP-free/20 mM $Na^+$-pipette solution induced an outward current in most $Kcnj11^{+/+}$ neurons recorded (*n* = 19 out of 26; mean for *n* = 26: 46.7 ± 19.0 pA at −50 mV, median value = 16.2 pA, $Chi^2$ = 5.538, p = 0.01860, one sample median test). In some neurons (*n* = 6 of 26), this procedure resulted in an outward current of more than 100 pA that reversed close to $E_K$ (see example in *Figure 3C, F*). In contrast, this current was not observed in $Kcnj11^{-/-}$ neurons ($U_{(26,22)}$ = 78, p = 2.4221 × 10$^{-6}$, one-tailed, *Figure 3D-G*). Instead, dialysis induced an inward current in most $Kcnj11^{-/-}$ neurons (*n* = 20 of 22; mean for *n* = 22: −59.9 ± 11.9 pA, *n* = 22, median value = −61.9 pA, $Chi^2$ = 14.727, p = 0.000124, one sample median test), suggesting that other conductances than the $K_{ATP}$ channels were also altered. Collectively, these data indicate that cortical neurons predominantly express functional $K_{ATP}$ channels composed of KCNJ11 and ABCC8 subunits.

## Modulation of neuronal excitability and activity by $K_{ATP}$ channel

Despite their large diversity, cortical neurons display a widespread functional expression of $K_{ATP}$ channels, questioning how these channels integrate the metabolic environment to adjust neuronal activity. To address this question, we first evaluated in identified cortical neurons (*n* = 39) the ability of

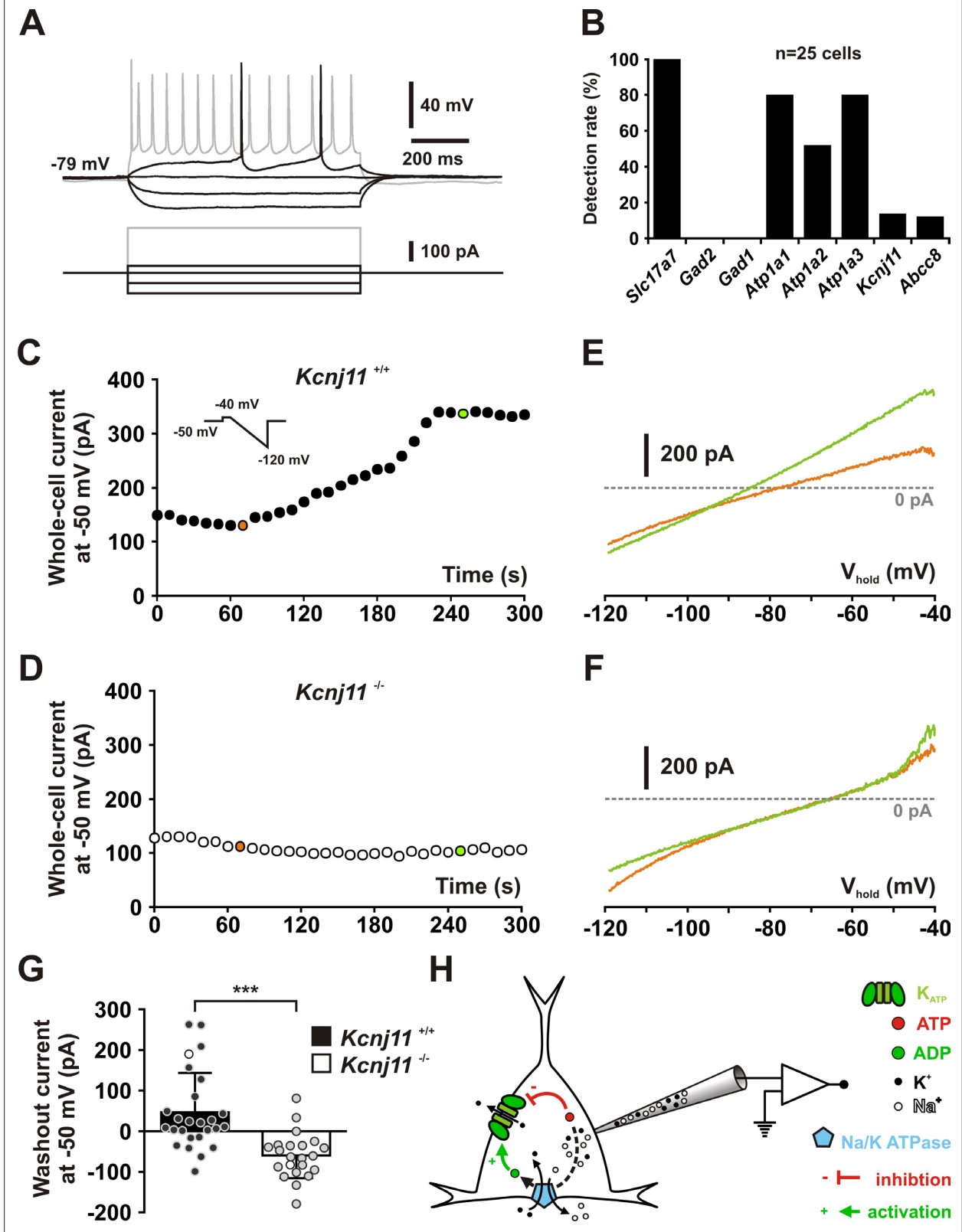

**Figure 3.** KCNJ11 is the pore-forming subunit of $K_{ATP}$ channels in cortical neurons. (**A**) Representative voltage responses of a mouse layer II/III regular spiking (RS) pyramidal cell induced by injection of current pulses (bottom traces). (**B**) Histograms summarizing the detection rate of *Slc17a7*, *Gad2* and 1, the *Atp1a1-3* subunits of the Na/K ATPase and the *Kcnj11* and *Abcc8* $K_{ATP}$ channel subunits in layer II/III regular spiking (RS) pyramidal cells from *Kcnj11*[+/+] mice. (**C, D**) Whole-cell stationary currents recorded at 50 mV during dialysis with ATP-free pipette solution in cortical neurons of *Kcnj11*[+/+] (**C**)

*Figure 3 continued on next page*

*Figure 3 continued*

and $Kcnj11^{-/-}$ (**D**) mice. Inset: voltage clamp protocol. (**E, F**) Current–voltage relationships obtained during ATP washout at the time indicated by green and orange circles in (**C, D**) in cortical neurons of $Kcnj11^{+/+}$ (**E**) and $Kcnj11^{-/-}$ (**F**) mice. (**G**) Histograms summarizing the whole-cell ATP washout currents in $Kcnj11^{+/+}$ (black) and $Kcnj11^{-/-}$ (white) cortical neurons. Data are expressed as mean ± standard error of the mean (SEM), and the individual data points are depicted. Open symbols in $Kcnj11^{+/+}$ and $Kcnj11^{-/-}$ bar plots indicate the cells illustrated in (**C, D**) and (**E, F**), respectively. (**H**) Diagram depicting the principle of the ATP washout experiment. *** indicates statistically significant with p< 0.001.

The online version of this article includes the following source data for figure 3:

**Source data 1.** Molecular profile of layer II–III pyramidal neurons shown in *Figure 3B*.

**Source data 2.** Statistical analysis of whole-cell ATP washout currents in $Kcnj11^{+/+}$ and $Kcnj11^{-/-}$ cortical neurons (shown in *Figure 3G*).

$K_{ATP}$ channels to modulate neuronal excitability, notably by measuring membrane potentials ($Q_{(2,39)}$ = 38.000, p = 5.6 × $10^{-9}$) and membrane resistances ($Q_{(2,39)}$ = 40.205, p = 1.9 × $10^{-9}$), as well as spiking activity ($Q_{(2,39)}$ = 28.593, p = 6.2 × $10^{-7}$). Following electrophysiological identification, the $K_{ATP}$ channel blocker tolbutamide was applied, which resulted in a slight depolarization ($\Delta V_m$ = 2.6 ± 0.8 mV, p = 1.74 × $10^{-2}$, *Figure 4A, B*) and increase in membrane resistance ($\Delta R_m$ = 78 ± 32 MΩ, p = 1.52 × $10^{-3}$, *Figure 4B, E*). These effects were strong enough to trigger and stimulate the firing of action potentials ($\Delta F$ = 0.3 ± 0.2 Hz, p = 9.21 × $10^{-3}$, *Figure 4A, C, F*). By contrast, diazoxide hyperpolarized cortical neurons (−4.0 ± 0.6 mV, p = 1.87 × $10^{-4}$, *Figure 4A, D*), decreased their membrane resistance (−39 ± 23 MΩ, p = 1.52 × $10^{-3}$, *Figure 4B, E*) but did alter their rather silent basal spiking activity (−0.1 ± 0.1 Hz, p = 0.821, *Figure 4A, C, F*).

Most cortical neurons (*n* = 32 of 39) showed modulation of neuronal excitability by both $K_{ATP}$ channel modulators and were considered to be responsive. A similar proportion of responsive neurons was observed between neuronal subtypes (*Figure 4—figure supplement 1A*, $Chi^2_{(5)}$ = 7.313, p = 0.1984) or groups (*Figure 4—figure supplement 1B*, p = 0.9999, Fisher's exact test). The apparent relative lack of responsiveness in FS-*Pvalb* interneurons (*Figure 4—figure supplement 1A*), despite a whole-cell $K_{ATP}$ conductance similar to that of other neuronal types, is likely attributable to their low input resistance (*Supplementary file 3*) making $K_{ATP}$ channels less effective to change membrane potential. Overall, $K_{ATP}$ channels modulated membrane potential, resistance, and firing rate by up to 7.9 ± 0.9 mV, 76 ± 17%, and 0.5 ± 0.2 Hz, respectively. This modulation of neuronal excitability (*Figure 4G–J*) and activity (*Figure 4—figure supplement 1C,D*) was similar between neuronal subtypes or groups (*Figure 4H–J* and *Figure 4—figure supplement 1C-E*). Thus, $K_{ATP}$ channels modulate the excitability and activity of all subtypes of cortical neurons.

## Enhancement of neuronal activity by lactate via modulation of $K_{ATP}$ channels

The expression of metabolically sensitive $K_{ATP}$ channels by cortical neurons suggests their ability to couple the local glycolysis capacity of astrocytes with spiking activity. We therefore evaluated whether extracellular changes in glucose and lactate could differentially shape the spiking activity of cortical neurons through their energy metabolism and $K_{ATP}$ channel modulation. Importantly, to preserve intracellular metabolism, neurons were recorded in perforated patch configuration. Stable firing rates of about 4 Hz inducing ATP consumption by the $Na^+/K^+$ ATPase (*Attwell and Laughlin, 2001*) were evoked by applying a depolarizing current and continuously monitored throughout changes in extracellular medium (*Figure 5A*, $Q_{(2,16)}$ = 22.625, p = 1.222 × $10^{-5}$).

Decreasing extracellular glucose from 10 mM to a normoglycemic concentration of 2.5 mM (*Silver and Erecińska, 1994*; *Hu and Wilson, 2002*) did not change firing rate (*Figure 5A and B*, p = 0.2159) of cortical neurons (*n* = 16). By contrast, supplementing extracellular 2.5 mM glucose with 15 mM lactate, an isoenergetic condition to 10 mM glucose for having the same number of carbon atoms, roughly doubled the firing rate compared to both 2.5 (p = 7.829 × $10^{-4}$) and 10 mM glucose (p = 4.303 × $10^{-6}$) conditions. Firing rate enhancement by lactate was dose dependent ($H_{(7,76)}$ = 35.142, p = 1.052 × $10^{-5}$) and reached statistical significance above 5 mM (*Figure 5C*). We reasoned that this effect could be mediated by $K_{ATP}$ channel closure. Indeed, the increase in firing rate by lactate (209 ± 49%) was strongly reduced by the $K_{ATP}$ channel activator diazoxide (71 ± 18%, p = 3.346 × $10^{-3}$, *Figure 5D*). Tolbutamide reversed diazoxide's effect (160 ± 17%, p = 9.345 × $10^{-3}$) but did not increase firing rate further (p = 0.5076). This occlusion of tolbutamide's effect by 15 mM lactate also suggests that this

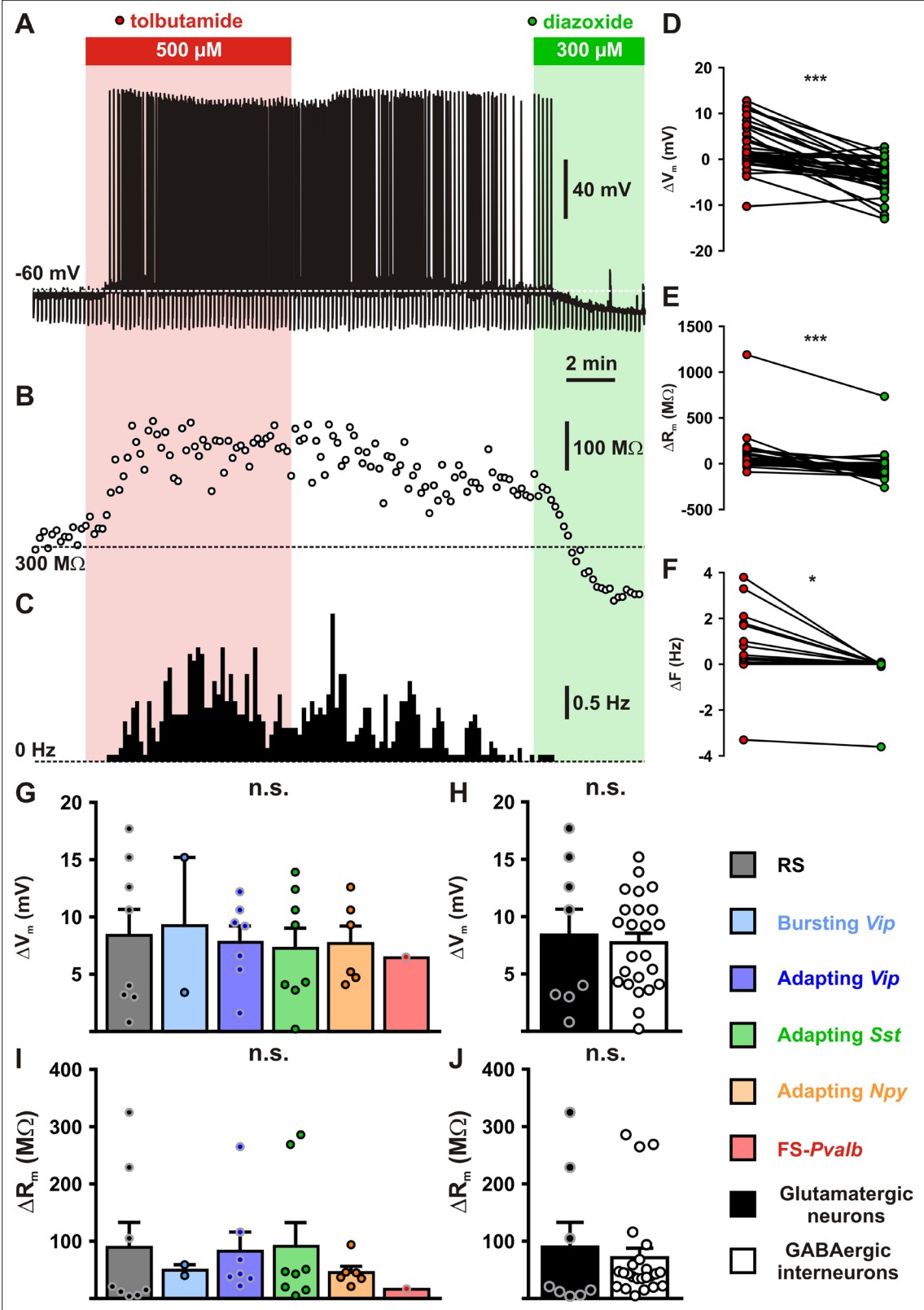

**Figure 4.** Modulation of cortical neuronal excitability and activity by $K_{ATP}$ channels. Representative example of a regular spiking (RS) neurons showing the changes in membrane potential (**A**), resistance (B, open circles) and spiking activity (**C**) induced by application of tolbutamide (red) and diazoxide (green). The colored bars and shaded zones indicate the application duration of $K_{ATP}$ channel modulators. Relative changes in membrane potential (**D**), resistance (**E**), and firing rate (**F**) induced by tolbutamide and diazoxide in cortical neurons. Histograms summarizing the modulation of membrane

*Figure 4 continued on next page*

*Figure 4 continued*

potential (G, $H_{(5,32)}$ = 0.15856, p = 0.999, and H, $U_{(8,24)}$ = 96, p = 1.0000) and resistance (I, $H_{(5,32)}$ = 2.7566, p = 0.737, and J, $U_{(8,24)}$ = 73, p = 0.3345) by $K_{ATP}$ channels in neuronal subtypes (**G, I**) and groups (**H, J**). Data are expressed as mean ± standard error of the mean (SEM), and the individual data points are depicted. n.s., not statistically significant. * and *** indicate statistically significant with p<0.05 and 0.001.

The online version of this article includes the following figure supplement(s) for figure 4:

**Source data 1.** Statistical analyses of membrane potential, membrane resistance and firing rate changes induced by $K_{ATP}$ channel modulators (shown in *Figure 4D, E*).

**Source data 2.** Statistical comparisons between neuronal subtypes and groups of the effect $K_{ATP}$ channel modulators on membrane potential, membrane resistance (shown in *Figure 4G–J*) and firing rate (shown in *Figure 4—figure supplement 1C,D*) as well as of the proportion of responsive neurons (shown in *Figure 4—figure supplement 1A,B*).

**Figure supplement 1.** Modulation of neuronal activity in different cortical neurons by $K_{ATP}$ channels.

concentration reaches saturating levels and is the highest metabolic state that can be sensed by $K_{ATP}$ channels. Enhancement of neuronal activity by lactate was also observed in *Kcnj11*$^{+/+}$ cortical neurons (147 ± 25%, p = 2.840 × 10$^{-2}$) but not in *Kcnj11*$^{-/-}$ mice (112 ± 32%, p = 0.8785, *Figure 5E*). These observations indicate that lactate enhances neuronal activity via a closure of $K_{ATP}$ channels (*Figure 5F*).

## Mechanism of lactate sensing

To determine whether lactate sensing involves intracellular lactate oxidative metabolism and/or extracellular activation of the lactate receptor GPR81, we next probed the expression of monocarboxylate transporters (MCTs), which allow lactate uptake. Consistent with mouse mRNAseq data (*Zeisel et al., 2015*; *Tasic et al., 2016*), *Slc16a1* (previously known as MCT1) and *Slc16a7* (previously known as MCT2) were the main transporters detected in rat cortical neurons, although with relatively low single-cell detection rates (54 of 277, 19.5 % and 78 of 277, 28.2%, for *Slc16a1* and *Slc16a7*, respectively, *Figure 6A* and *Figure 6—figure supplement 1*).

The expression of MCTs in cortical neurons is compatible with lactate uptake and metabolism leading to the closure of $K_{ATP}$ channels and an increase in firing rate. We thus evaluated whether lactate uptake was needed for lactate sensing. We used 250 µM α-cyano-4-hydroxycinnamic acid (4-CIN), a concentration blocking lactate uptake while only moderately altering mitochondrial pyruvate carrier in brain slices (*Schurr et al., 1999*; *Ogawa et al., 2005*; *Galeffi et al., 2007*). 4-CIN reversed the increased firing rate induced by lactate (*Figure 6B*, T(9) = 0, p = 7.686 × 10$^{-3}$) indicating that facilitated lactate transport is required for $K_{ATP}$ channel closure and in turn firing rate acceleration.

A mechanism of lactate sensing involving an intracellular lactate oxidative metabolism would also require the expression of lactate dehydrogenase (LDH), that reversibly converts lactate and nicotinamide adenine dinucleotide (NAD$^+$) to pyruvate and NADH (*Figure 6E*, inset). We thus also probed for the expression of *Ldh* subunits. *Ldha* and *Ldhb* were observed in a large majority of cortical neurons with *Ldha* being more frequent in glutamatergic neurons than in GABAergic interneurons (p = 1.61 × 10$^{-2}$, *Figure 6A* and *Figure 6—figure supplement 1*). Nonetheless, neuron subtypes analysis did not allow to disclose which populations express less frequently *Ldha* (*Figure 6—figure supplement 1*). To confirm the ability of cortical neurons to take up and oxidize lactate we also visualized NADH fluorescence dynamics (*Chance et al., 1962*) induced by bath application of lactate. Widefield somatic NADH fluorescence appeared as a diffuse labeling surrounding presumptive nuclei (*Figure 6D*). Consistent with lactate transport by MCTs and oxidization by LDH, NADH was increased under lactate application ($U_{(61,67)}$ = 196, p = 3.1 × 10$^{-24}$, *Figure 6E, F*).

Since the lactate receptor GPR81 has been observed in the cerebral cortex (*Lauritzen et al., 2014*), lactate sensing might also involve this receptor. This possibility was ruled out by the observation that pyruvate (15 mM), which is transported by MCTs (*Bröer et al., 1998*; *Bröer et al., 1999*) but does not activate GPR81 (*Ahmed et al., 2010*), enhanced firing rate to an extent similar to that of lactate (*Figure 6C*, $U_{(16,6)}$ = 43, p = 0.7468). In line with its uptake and reduction, pyruvate also decreased NADH (*Figure 6E, F*, $U_{(44,67)}$=868, p = 2.08 × 10$^{-4}$).

The requirement of monocarboxylate transport and the similar effect of lactate and pyruvate on neuronal activity suggest that once taken up, lactate would be oxidized into pyruvate and metabolized by mitochondria to produce ATP, leading in turn to a closure of $K_{ATP}$ channels and increased firing rate. The apparent absence of glucose responsiveness in cortical neurons also suggests that

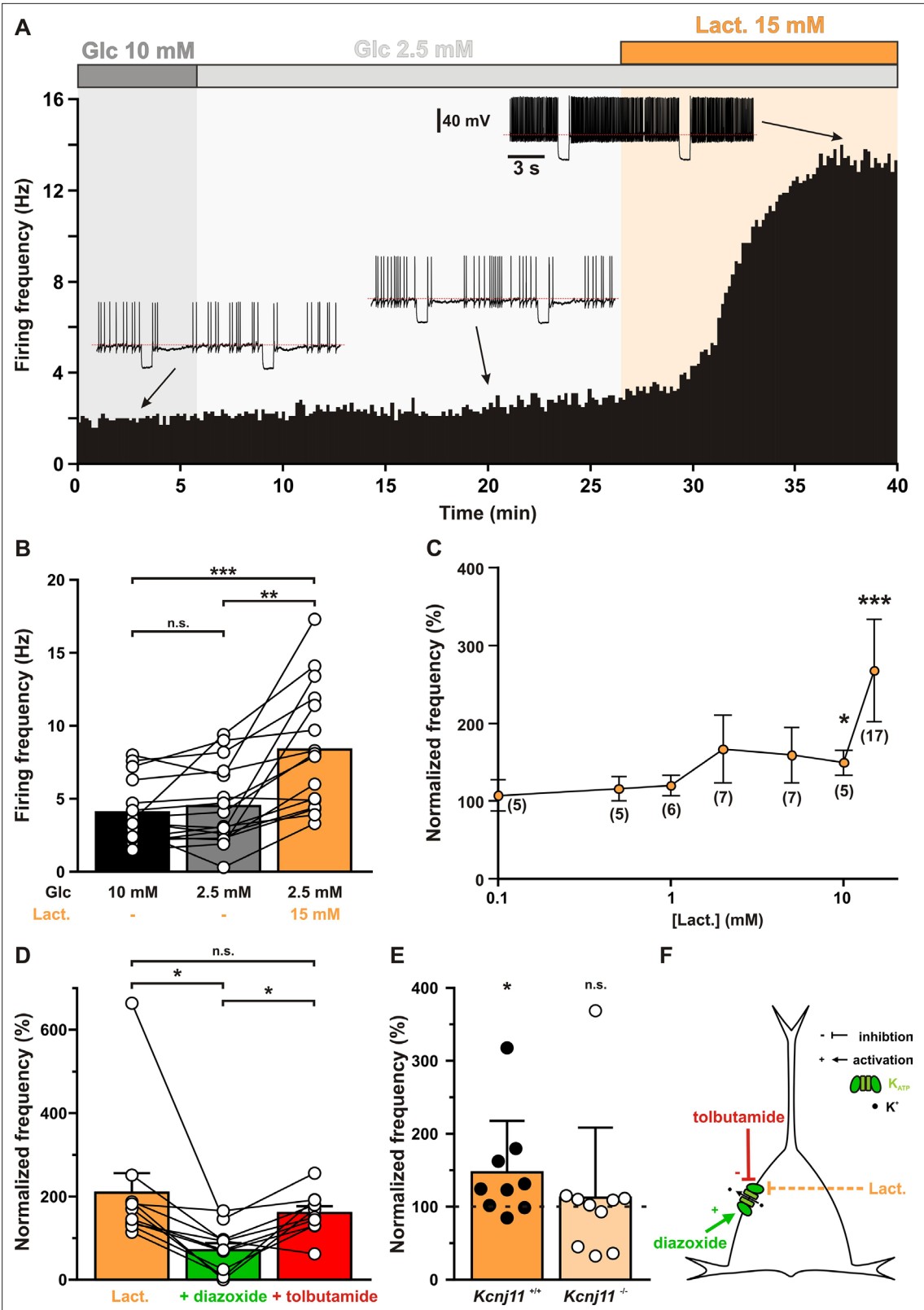

**Figure 5.** Lactate enhances cortical neuronal activity via $K_{ATP}$ channel modulation. (**A**) Representative perforated patch recording of an adapting vasoactive intestinal polypeptide (VIP) neuron showing the modulation of firing frequency induced by changes in the extracellular concentrations of metabolites. The colored bars and shaded zones indicate the concentration in glucose (gray) and lactate (orange). Voltage responses recorded at the time indicated by arrows. The red dashed lines indicate −40 mV. (**B**) Histograms summarizing the mean firing frequency during changes in extracellular

*Figure 5 continued on next page*

*Figure 5 continued*

concentration of glucose (black and gray) and lactate (orange). Data are expressed as mean ± standard error of the mean (SEM), and the individual data points are depicted. n.s., not statistically significant. *, ** and *** indicate statistically significant with p< 0.05, 0.01 and 0.001, respectively. (**C**) Dose-dependent enhancement of firing frequency by lactate. Data are normalized by the mean firing frequency in absence of lactate and are expressed as mean ± SEM. Numbers in brackets indicate the number of recorded neurons at different lactate concentrations. (**D**) Histograms summarizing the normalized frequency under 15 mM lactate (orange) and its modulation by addition of diazoxide (green) or tolbutamide (red). Data are expressed as mean ± SEM, and the individual data points are depicted. n.s., not statistically significant. (**E**) Histograms summarizing the enhancement of normalized frequency by 15 mM lactate in *Kcnj11⁺/⁺* (orange) and *Kcnj11⁻/⁻* (pale orange) mouse cortical neurons. The dash line indicates the normalized mean firing frequency in absence of lactate. Data are expressed as mean ± SEM, and the individual data points are depicted. (**F**) Diagram depicting the enhancement of neuronal activity by lactate via modulation of $K_{ATP}$ channels.

The online version of this article includes the following figure supplement(s) for figure 5:

**Source data 1.** Statistical analysis of the effect of glucose and lactate on firing rate (shown in *Figure 5B*).

**Source data 2.** Statistical analysis of dose-dependent enhancement of firing frequency by lactate (shown in *Figure 5C*).

**Source data 3.** Statistical analysis of the effect of diazoxide and tolbutamide on firing rate enhancement by lactate (shown in *Figure 5D*).

**Source data 4.** Statistical comparison of lactate enhancement of normalized frequency in Kcnj11⁺/⁺ and Kcnj11⁻/⁻ (shown in *Figure 5E*).

glycolysis contributes modestly to ATP production. To determine the relative contribution of glycolysis and oxidative phosphorylation to ATP synthesis, we transduced the genetically encoded fluorescence resonance energy transfer (FRET)-based ATP biosensor AT1.03$^{YEMK}$ (*Imamura et al., 2009*) using a recombinant Sindbis virus. AT1.03$^{YEMK}$ fluorescence was mostly observed in pyramidal shaped cells (*Figure 6G*), consistent with the strong tropism of this viral vector toward pyramidal neurons (*Piquet et al., 2018*). Blocking glycolysis with 200 µM iodoacetic acid (IAA) decreased modestly the FRET ratio by 2.9 ± 0.2% (*Figure 6H*, p = 2.44 × 10⁻¹³). By contrast, adding potassium cyanide (KCN, 1 mM), a respiratory chain blocker, reduced the FRET ratio to a much larger extent (52.3 ± 0.6%, *Figure 6H*, p = 2.44 × 10⁻¹³). KCN also induced a strong NADH fluorescence increase (*Figure 6—figure supplement 2A, B*, $U_{(12,42)}$=0, p = 5.83 × 10⁻¹²), indicating a highly active oxidative phosphorylation in cortical neurons.

## Discussion

We report that in juvenile rodents extracellular lactate and pyruvate, but not glucose, enhance the activity of cortical neurons through a mechanism involving facilitated transport and the subsequent closure of $K_{ATP}$ channels composed of KCNJ11 and ABCC8 subunits. ATP synthesis derives mostly from oxidative phosphorylation and weakly from glycolysis in cortical neurons. Together with their ability to oxidize lactate by LDH, these observations suggest that lactate is a preferred energy substrate over glucose in cortical neurons. Besides its metabolic importance lactate also appears as a signaling molecule enhancing firing activity (*Figure 7*). This suggests that an efficient neurovascular and neurometabolic coupling could define a time window of an up state of lactate during which neuronal activity and plasticity would be locally enhanced (*Suzuki et al., 2011*; *Jimenez-Blasco et al., 2020*).

### $K_{ATP}$ channel subunits in cortical neurons

Similar to neurons of the hippocampal formation (*Zawar et al., 1999*; *Cunningham et al., 2006*; *Sada et al., 2015*) we found that, regardless of the neuronal type, most neocortical neurons express diazoxide-sensitive, but pinacidil-insensitive $K_{ATP}$ channels (*Cao et al., 2009*). Since $K_{ATP}$ channel modulators were bath applied, the induced currents recorded from individual cells could also reflect network interactions with neurons and/or astrocytes expressing $K_{ATP}$ channels (*Thomzig et al., 2001*; *Matsumoto et al., 2002*). However, the kinetics and reversal potential of the steady-state outward currents evoked by $K_{ATP}$ channel modulations do not support an indirect effect induced by transmitter release. In agreement with the observed pharmacological profile (*Inagaki et al., 1996*) and the absence of functional $K_{ATP}$ channels in *Kcnj11⁻/⁻* neurons, we observed that *Kcnj11* and *Abcc8* subunits were the main components of $K_{ATP}$ channels as detected by ribo-tag-based transcriptomics for many neuronal types (*Doyle et al., 2008*).

Their low detection rate by scRT-PCR is presumably due to the low copy number of their mRNAs, to the low RT efficiency and to the harvesting procedure restricted to the soma. Indeed, a single-cell RNAseq study performed in mouse somatosensory cortex (*Zeisel et al., 2015*) revealed about 5

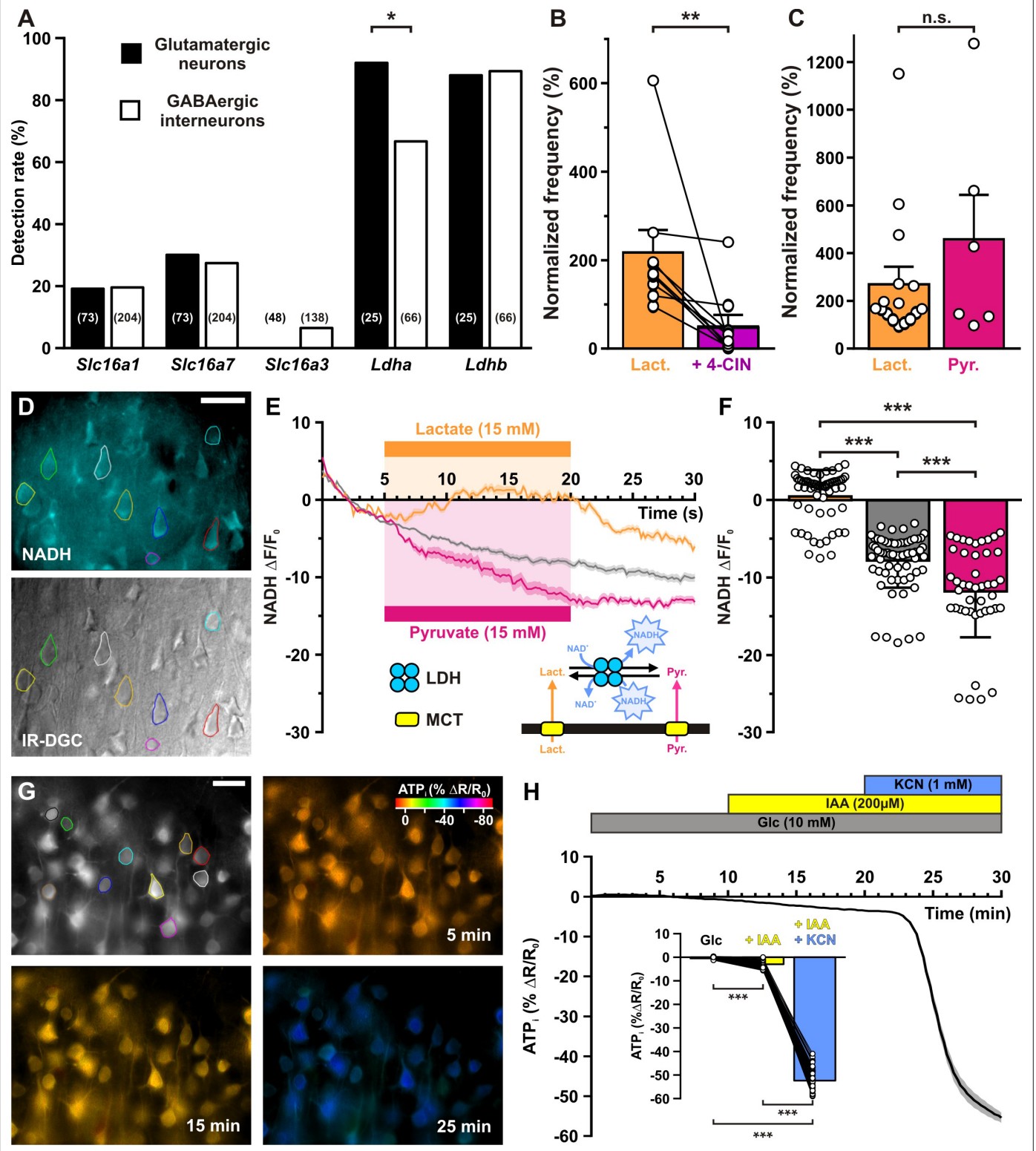

**Figure 6.** Lactate enhancement of cortical neuronal activity involves lactate uptake and metabolism. (**A**) Histograms summarizing the detection rate of the monocarboxylate transporters *Slc16a1*, *7*, and 3 and *Ldha* and *b* lactate dehydrogenase subunits in glutamatergic neurons (black) and GABAergic interneurons (white). The numbers in brackets indicate the number of analyzed cells. (**B**) Histograms summarizing the enhancement of normalized frequency by 15 mM lactate (orange) and its suppression by the monocarboxylate transporters (MCTs) inhibitor α-cyano-4-hydroxycinnamic acid (4-CIN; purple). Data are expressed as mean ± standard error of the mean (SEM), and the individual data points are depicted. (**C**) Histograms summarizing

*Figure 6 continued on next page*

*Figure 6 continued*

the enhancement of normalized frequency by 15 mM lactate (orange) and pyruvate (magenta). Data are expressed as mean ± SEM, and the individual data points are depicted n.s., not statistically significant. (**D**) Widefield NADH (reduced form of nicotinamide adenine dinucleotide) autofluorescence (upper panel, scale bar: 20 μm) and corresponding field of view observed under IR-DGC (lower panel). The somatic regions of interest are delineated. (**E**) Mean relative changes in NADH autofluorescence in control condition (gray) and in response to 15 mM lactate (orange) or pyruvate (magenta). The colored bars indicate the duration of applications. Data are expressed as mean ± SEM. Inset: diagram depicting the NADH changes induced by lactate and pyruvate uptake by MCT and their interconversion by lactate dehydrogenase (LDH). (**F**) Histograms summarizing the mean relative changes in NADH autofluorescence measured during the last 5 min of 15 mM lactate (orange) or pyruvate (magenta) application and corresponding time in control condition (gray). Data are expressed as mean ± SEM, and the individual data points are depicted. (**G**) Widefield YFP fluorescence of the ATP biosensor AT1.03$^{YEMK}$ (upper left panel, scale bar: 30 μm) and pseudocolor images showing the intracellular ATP (YFP/CFP ratio value coded by pixel hue, see scale bar in upper right panel) and the fluorescence intensity (coded by pixel intensity) at different times under 10 mM extracellular glucose (upper right panel) and after addition of iodoacetic acid (IAA; lower left panel) and potassium cyanide (KCN; lower right panel). (**H**) Mean relative changes in intracellular ATP (relative YFP/CFP ratio) measured under 10 mM extracellular glucose (gray) and after addition of IAA (yellow) and KCN (blue). Data are expressed as mean ± SEM. The colored bars indicate the time and duration of metabolic inhibitor application. Inset: Histograms summarizing the mean relative changes in intracellular ATP (relative YFP/CFP ratio) ratio under 10 mM extracellular glucose (gray) and after addition of IAA (yellow) and KCN (blue). Data are expressed as mean ± SEM, and the individual data points are depicted. *, ** and *** indicate statistically significant with p< 0.05, 0.05 and 0.001, respectively.

The online version of this article includes the following source data and figure supplement(s) for figure 6:

**Source data 1.** Statistcal comparisons of the detection rate of monocarboxylate transporters and lactate dehydrogenase subunits between neuronal groups (shown in *Figure 6A*) and subtypes (shown in *Figure 6—figure supplement 1*).

**Source data 2.** Statistical analysis of the effect of monocarboxylate transporter (MCT) inhibition by α-cyano-4-hydroxycinnamic acid (4-CIN) on lactate-enhanced firing rate (shown in *Figure 6B*).

**Source data 3.** Statistcal comparison of the relative effect of lactate and pyruvate on firing rate enhancement (shown in *Figure 6C*).

**Source data 4.** Statistcal comparisons of the relative effects of lactate, pyruvate, and control condition on the the mean relative changes in NADH autofluorescence (shown in *Figure 6F*).

**Source data 5.** Satistical analysis of the effects of iodoacetic acid (IAA) and potassium cyanide (KCN) on the relative changes in intracellular ATP (shown in *Figure 6H* inset).

**Figure supplement 1.** Detection rate of monocarboxylate transporters and lactate dehydrogenase subunits in different cortical neuronal types.

**Figure supplement 2.** Neuronal NADH autofluorescence increase by blockade of oxidative phosphorylation.

**Figure supplement 2—source data 1.** Statistcal analysis of effect of potassium cyanide (KCN) on the mean relative changes in NADH autofluorescence (shown in *Figure 6—figure supplement 2*).

molecules of both *Kcnj11* and *Abcc8* mRNAs per cell in cortical neurons, whereas scRT-PCR detection limit was estimated to be around 25 molecules of mRNA in the patch pipette (*Tsuzuki et al., 2001*). Furthermore, since *Kcnj11* is an intronless gene, collection of the nucleus was avoided to prevent potential false positives. Thus, neurons positive for both *Kcnj11* and *Sst* intron, taken as an indicator of genomic DNA (*Hill et al., 2007*; *Devienne et al., 2018*), were discarded from *Kcnj11* expression analysis. Unavoidably, this procedure does reduce the amount of cytoplasm collected, thereby decreasing the detection rate of both *Kcnj11* and *Abcc8*.

Consistent with the preferred expression of *Kcnj8* in mural and endothelial cells (*Bondjers et al., 2006*; *Zeisel et al., 2015*; *Tasic et al., 2016*; *Aziz et al., 2017*; *Vanlandewijck et al., 2018 Saunders et al., 2018*), this subunit was only observed in one out of 277 cortical neurons analyzed. Similarly, SUR2B, the *Abcc9* variant expressed in forebrain (*Isomoto et al., 1996*) and cortex (*Figure 1—figure supplement 1B*), whose presence is largely restricted to vascular cells (*Zeisel et al., 2015*), was not observed in cortical neurons.

## Relative sensitivity of cortical neurons to glucose, lactate, and pyruvate

Consistent with previous observations (*Yang et al., 1999*), decreasing extracellular glucose from standard slice concentrations down to a normoglycemic level did not alter firing rates of cortical neurons. However, their activity is silenced during hypoglycemic episodes through K$_{ATP}$ channels activation (*Yang et al., 1999*; *Zawar and Neumcke, 2000*; *Molnár et al., 2014*; *Sada et al., 2015*). This relative glucose unresponsiveness is in contrast with pancreatic beta cells and hypothalamic glucose-excited neurons whose activity is regulated over a wider range of glucose concentrations by K$_{ATP}$ channels also composed with KCNJ11 and ABCC8 subunits (*Aguilar-Bryan et al., 1995Inagaki et al., 1995a*; *Miki et al., 1998*; *Yang et al., 1999*; *Miki et al., 2001*; *Tarasov et al., 2006*; *Varin et al., 2015* ). The

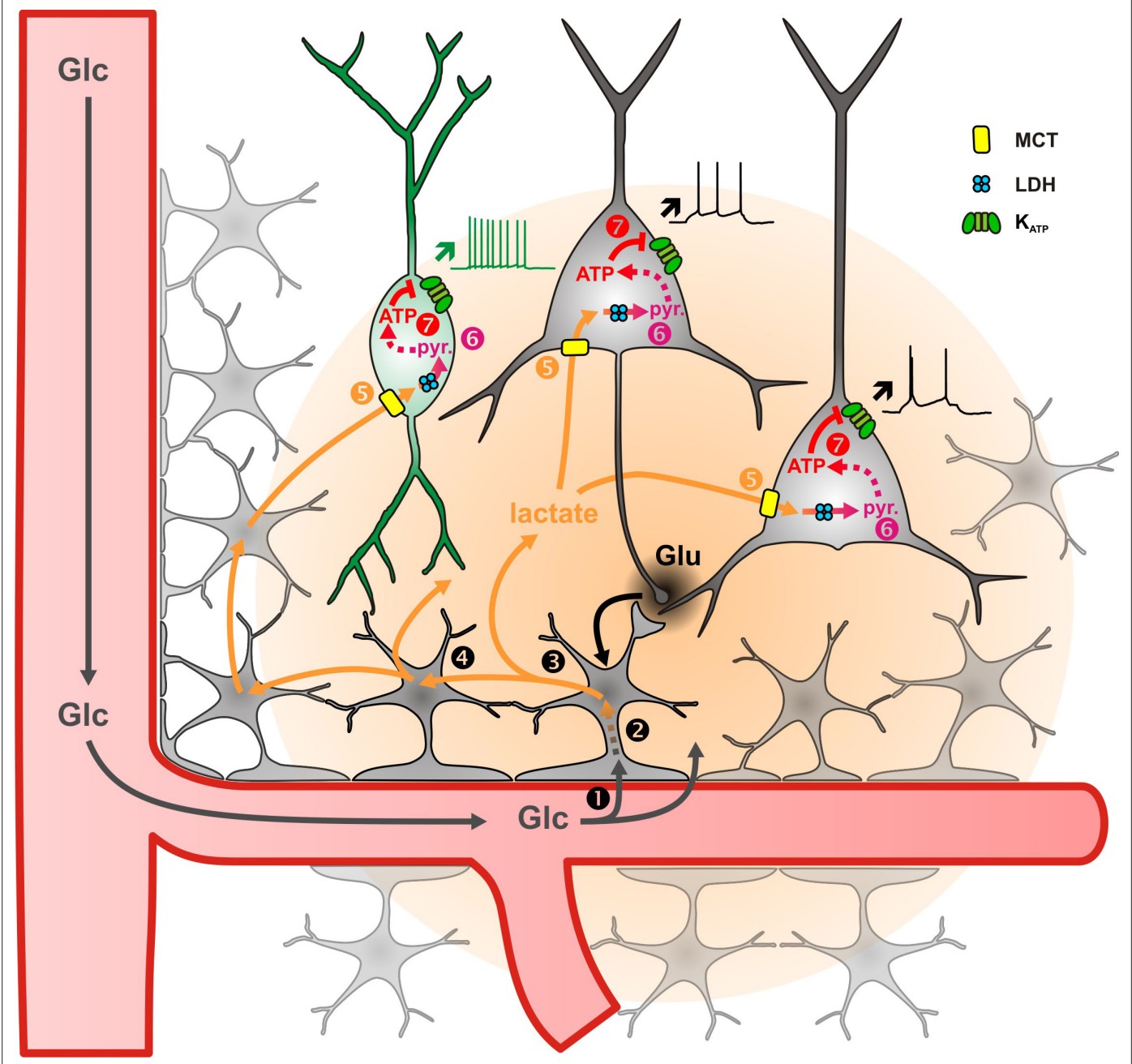

**Figure 7.** Diagram summarizing the mechanism of lactate sensing in the cortical network. Glutamate (Glu) released during synaptic transmission stimulates (1) blood glucose (Glc) uptake in astrocytes, (2) aerobic glycolysis, (3) lactate release, and (4) diffusion through the astrocytic network. Lactate is then (5) taken up by neurons via monobarboxylate transporters (MCT) and (6) oxidized into pyruvate by lactate dehydrogenase (LDH). The ATP produced by pyruvate oxidative metabolism (7) closes KATP channels and increases the spiking activity of both pyramidal cells (black) and inhibitory interneurons (green). The color gradient of the circles represents the extent of glutamate (black) and lactate (orange) diffusion, respectively. Dashed arrows indicate multisteps reactions.

inability of cortical neurons to regulate their spiking activity at glucose levels beyond normoglycemia is likely due to the lack of glucokinase, a hexokinase which catalyzes the first step of glycolysis and acts as a glucose sensor in the millimolar range (*German, 1993*; *Yang et al., 1999*). As earlier reported, hexokinase-1 (*Hk1*) is the major isoform in cortical neurons (*Zeisel et al., 2015*; *Tasic et al., 2016*; *Piquet et al., 2018*). Since this enzyme has a micromolar affinity for glucose and is inhibited by its product, glucose-6-phosphate (*Wilson, 2003*), HK1 is likely already saturated and/or inhibited during

normoglycemia thereby limiting glycolysis. Nonetheless, HK1 saturation/inhibition can be mitigated when energy consumption is high (**Attwell and Laughlin, 2001**; **Wilson, 2003**; **Tantama et al., 2013**), and then glucose can probably modulate neuronal activity via a high-affinity mechanism, as evidenced by slow oscillations of spiking activity involving synaptic transmission (**Cunningham et al., 2006**) or by the use of glucose-free whole-cell patch-clamp solution (**Kawamura et al., 2010**) that mimics high glucose consumption (**Piquet et al., 2018**; **Díaz-García et al., 2019**).

Similar to glucose-excited hypothalamic neurons (**Yang et al., 1999**; **Song and Routh, 2005**), but in contrast with pancreatic beta cells (**Newgard, 1995**), cortical neurons were dose dependently excited by lactate. This lactate sensitivity is consistent with lactate transport and oxidization in hypothalamic and cortical neurons (**Ainscow et al., 2002**; **Sada et al., 2015**; **Díaz-García et al., 2017**) which are low in beta cells (**Sekine et al., 1994**; **Pullen et al., 2011**). Pyruvate had a similar effect to lactate in cortical neurons under normoglycemic condition whereas it only maintains the activity of hypothalamic glucose-excited neurons during hypoglycemia (**Yang et al., 1999**) and barely activates pancreatic beta cells (**Düfer et al., 2002**). Thus, cortical neurons display a peculiar metabolic sensitivity to monocarboxylates. Our data also suggest that under normoglycemic conditions a portion of $K_{ATP}$ channels are open when cortical neurons fire action potentials.

## Mechanism of lactate sensing

Our pharmacological, molecular, and genetic evidence indicates that the closure of $K_{ATP}$ channels is responsible for the firing rate enhancement by lactate. Since $K_{ATP}$ channels can be modulated by G-protein-coupled receptors (**Kawamura et al., 2010**), lactate sensing might have been mediated by GPR81, a $G_i$-protein-coupled lactate receptor expressed in the cerebral cortex (**Lauritzen et al., 2014**). This possibility is however unlikely since the activation of GPR81 inhibits cultured cortical neurons (**Bozzo et al., 2013**; **de Castro Abrantes et al., 2019**) and we show here that enhancing effect pyruvate on neuronal activity was similar to that of lactate, although pyruvate does not activate GPR81 (**Ahmed et al., 2010**).

We found that lactate sensing was critically dependent on lactate transport and we confirmed the capacity of cortical neurons to take up and oxidize lactate (**Bittar et al., 1996**; **Laughton et al., 2000**; **Bouzier-Sore et al., 2003**; **Wyss et al., 2011**; **Choi et al., 2012**; **Sada et al., 2015**; **Mächler et al., 2016**). Although *Slc16a1* and *Slc16a7* mRNAs were infrequently detected by scRT-PCR, our imaging and electrophysiological observations indicate a widespread transport of lactate. Similar to $K_{ATP}$ channel subunits, the relatively low single-cell detection rates are likely due to the low copy number of both *Slc16a1* and *Slc16a7* mRNAs which have been reported to be less than 10 copies per cell in cortical neurons (**Zeisel et al., 2015**). Interestingly, discrepancies between mRNA and protein expression have been reported for MCTs (**Pierre and Pellerin, 2005**) which may reflect regulation at the translational level and/or a low turnover of the proteins. The ability of cortical neurons to oxidize lactate is supported by both scRT-PCR and NADH imaging observations. The much higher detection rates of *Ldha* and *Ldhb* mRNA parallel their single-cell copy number which is two to five times higher than that of *Ldha* and *Ldhb* (**Zeisel et al., 2015**).

The impairment of lactate-enhanced firing by 4-CIN might be due to the blockade of lactate uptake by neurons but also to the blockade of lactate efflux by astrocytes. However, it is unlikely that astrocytes have a substantial contribution here. First, basal lactate tone in cortical slices has been estimated to be about 200 µM (**Karagiannis et al., 2016**), a concentration with little or no effect on lactate sensing (**Figure 5C**). Second, in addition to MCTs, astrocytes can also release lactate from connexin hemichannels (**Karagiannis et al., 2016**) and from a lactate-permeable ion channel (**Sotelo-Hitschfeld et al., 2015**). Hence, blockade of MCTs by 4-CIN would have, at most, only partially altered the release of lactate by astrocytes.

LDH metabolites, including pyruvate and oxaloacetate, can lead to $K_{ATP}$ channel closure (**Dhar-Chowdhury et al., 2005**; **Sada et al., 2015**) and could mediate lactate sensing. An intermediate role of oxaloacetate in lactate sensing is compatible with enhanced Krebs cycle and oxidative phosphorylation, which leads to an increased ATP/ADP ratio and the closure of $K_{ATP}$ channels (**Figure 7**). In contrast to oxaloacetate, intracellular ATP was found to be ineffective for reverting $K_{ATP}$ channel opening induced by LDH inhibition (**Sada et al., 2015**). Interestingly, hippocampal interneurons were found to be insensitive to glucose deprivation in whole-cell configuration (**Sada et al., 2015**) but not in perforated patch configuration (**Zawar and Neumcke, 2000**) whereas almost the opposite

was found in CA1 pyramidal cells. Whether altered intracellular metabolism by whole-cell recording accounted for the apparent lack of ATP sensitivity remains to be determined.

Increased firing rate by lactate metabolism is likely to enhance sodium influx and stimulate ATP comsumption by the Na$^+$/K$^+$ ATPase (*Tanner et al., 2011*). This could in turn lower ATP/ADP ratio, increase the $p_0$ of K$_{ATP}$ channels (*Tantama et al., 2013*) and subsequently decrease firing rate. We did not observe such a decrease and, once firing rate was enhanced, it remained stable for several minutes (*Figure 5A*). This suggests that ATP levels remained relatively stable, as reported in pancreatic cells under high glucose stimulation that recruits calcium-dependent energy metabolism (*Tanaka et al., 2014*). However, when energy consumption is high, as during network synaptic transmission, fluctuations of ATP/ADP ratio and slow oscillations of spiking activity can occur as observed in the entorhinal cortex (*Cunningham et al., 2006*).

## Lactate as an energy substrate for neurons and an enhancer of spiking activity and neuronal plasticity

We confirmed that the ATP produced by cortical neurons was mostly derived from oxidative phosphorylation and marginally from glycolysis (*Almeida et al., 2001*; *Hall et al., 2012*). Together with the enhancement of spiking activity through K$_{ATP}$ channels by lactate, but not by glucose, our data support both the notion that lactate is a preferred energy substrate over glucose for neonatal and juvenile cortical neurons (*Bouzier-Sore et al., 2003Ivanov et al., 2011*) as well as the astrocyte–neuron lactate shuttle hypothesis (*Pellerin and Magistretti, 1994*). Whether lactate sensing persists in the adult remains to be determined.

Although the local cellular origin of lactate has been recently questioned (*Lee et al., 2012*; *Díaz-García et al., 2017*), a growing number of evidence indicates that astrocytes are major central lactate producers (*Almeida et al., 2001*; *Choi et al., 2012*; *Sotelo-Hitschfeld et al., 2015*; *Karagiannis et al., 2016*; *Le Douce et al., 2020*; *Jimenez-Blasco et al., 2020*).

Glutamatergic synaptic transmission stimulates blood glucose uptake, astrocyte glycolysis, as well as lactate release (*Pellerin and Magistretti, 1994*; *Voutsinos-Porche et al., 2003*; *Ruminot et al., 2011*; *Choi et al., 2012*; *Sotelo-Hitschfeld et al., 2015*; *Lerchundi et al., 2015*) and diffusion through the astroglial gap junctional network (*Rouach et al., 2008*). This indicates that local and fast glutamatergic synaptic activity would be translated by astrocyte metabolism into a widespread and long-lasting extracellular lactate increase (*Prichard et al., 1991*; *Hu and Wilson, 1997*), which could in turn enhance the firing of both excitatory and inhibitory neurons (*Figure 7*). Such a lactate surge would be spatially confined by the gap junctionnal connectivity of the astroglial network, which in layer IV represents an entire barrel (*Houades et al., 2008*).

This suggests that increased astrocytic lactate induced by whisker stimulation could enhance the activity of the cortical network and fine-tune upcoming sensory processing for several minutes, thereby favoring neuronal plasticity. Along this line, lactate derived from astrocyte glycogen supports both neuronal activity and long-term memory formation (*Suzuki et al., 2011*; *Choi et al., 2012*; *Vezzoli et al., 2020*). Similarly, cannabinoids, which notably alter neuronal processing and memory formation (*Stella et al., 1997*), hamper lactate production by astrocytes (*Jimenez-Blasco et al., 2020*).

In contrast to glucose levels, lactate levels are higher in extracellular fluid than in plasma and can be as high as 5 mM under basal resting condition (*Abi-Saab et al., 2002*; *Zilberter et al., 2010*). Given that extracellular lactate is almost doubled during neuronal activity (*Prichard et al., 1991*; *Hu and Wilson, 1997*), enhancement of neuronal activity by lactate is likely to occur when the brain is active. Peripheral lactate released by skeletal muscles, which can reach 15 mM in plasma following an intense physical exercise (*Quistorff et al., 2008*), could also facilitate this effect. Although systemic increase of lactate elevates its cerebral extracellular concentration (*Mächler et al., 2016*; *Carrard et al., 2018*) to a level with little or no effect on firing rate, when both the brain and the body are active, as during physical exercise, both astrocytes and systemic lactate could contribute to enhance spiking activity.

Blood-borne lactate has been shown to promote learning and memory formation via brain-derived neurotrophic factor (*El Hayek et al., 2019*). It is worth noting that the production of this neurotrophin is altered in *Kcnj11*$^{-/-}$ mice and impaired by a K$_{ATP}$ channel opener (*Fan et al., 2016*), both conditions compromising the effect of lactate on spiking activity. Hence, the increase in astrocyte and systemic lactate could fine-tune neuronal processing and plasticity in a context-dependent manner and their coincidence could be potentially synergistic.

## Lactate-sensing compensatory mechanisms

Since excitatory neuronal activity increases extracellular lactate (*Prichard et al., 1991*; *Hu and Wilson, 1997*) and lactate enhances neuronal activity, such a positive feedback loop (*Figure 7*) suggests that compensatory mechanisms might be recruited to prevent an overexcitation of neuronal activity by lactate supply. A metabolic negative feedback mechanism could involve the impairment of astrocyte metabolism and lactate release by endocannabinoids (*Jimenez-Blasco et al., 2020*) produced during intense neuronal activity (*Stella et al., 1997*).

Another possibility would consist in a blood flow decrease that would in turn reduce the delivery of blood glucose and subsequent local lactate production and release but also blood-borne lactate. Some GABAergic interneuron subtypes (*Cauli et al., 2004*; *Uhlirova et al., 2016*; *Krawchuk et al., 2019*), but also astrocytes (*Girouard et al., 2010*), can trigger vasoconstriction and blood flow decrease when their activity is increased. This could provide a negative hemodynamic feedback restricting spatially and temporally the increase of spiking activity by lactate.

PVALB- and SST-expressing interneurons exhibit higher mitochondrial content and apparent oxidative phosphorylation than pyramidal cells (*Gulyás et al., 2006*) suggesting that interneurons would more rapidly metabolize and sense lactate than pyramidal cells. These inhibitory GABAergic interneurons might therefore silence the cortical network, thereby providing a negative neuronal feedback loop. Active decrease in blood flow is associated with a decrease in neuronal activity (*Shmuel et al., 2002*; *Shmuel et al., 2006*; *Devor et al., 2007*). Vasoconstrictive GABAergic interneurons may underlie for both processes and could contribute to returning the system to a low lactate state.

## Conclusion

Our data indicate that lactate is both an energy substrate for cortical neurons and a signaling molecule enhancing their spiking activity. This suggests that a coordinated neurovascular and neurometabolic coupling would define a time window of an up state of lactate that, besides providing energy and maintenance to the cortical network, would fine-tune neuronal processing and favor, for example, memory formation (*Suzuki et al., 2011*; *Kann et al., 2014*; *Galow et al., 2014*; *Jimenez-Blasco et al., 2020*).

# Materials and methods

## Lead contact and materials availability

Further information and requests for resources and reagents should be directed to, and will be fulfilled by, the lead contact, B. Cauli (bruno.cauli@upmc.fr).

## Experimental model and subject details

Wistar rats, C57BL/6RJ or *Kcnj11*$^{-/-}$ (B6.129P2-*Kcnj11*$^{tm1Sse}$, backcrossed into C57BL6 over six generations) mice were used for all experiments in accordance with French regulations (Code Rural R214/87 to R214/130) and conformed to the ethical guidelines of both the directive 2010/63/EU of the European Parliament and of the Council and the French National Charter on the ethics of animal experimentation. A maximum of three rats or five mice were housed per cage and single animal housing was avoided. Male rats and mice of both genders were housed on a 12 hr light/dark cycle in a temperature-controlled (21–25°C) room and were given food and water ad libitum. Animals were used for experimentation at 13–24 days of age.

## Cortical slice preparation

Rats or mice were deeply anesthetized with isoflurane. After decapitation brains were quickly removed and placed into cold (~4 °C) oxygenated artificial cerebrospinal fluid (aCSF) containing (in mM): 126 NaCl, 2.5 KCl, 1.25 $NaH_2PO_4$, 2 $CaCl_2$, 1 $MgCl_2$, 26 $NaHCO_3$, 10 glucose, 15 sucrose, and one kynurenic acid. Coronal slices (300 μm thick) containing the barrel cortex were cut with a vibratome (VT1000S, Leica) and allowed to recover at room temperature for at least 1 hr in aCSF saturated with $O_2/CO_2$ (95%/5%) as previously described (*Karagiannis et al., 2009*; *Devienne et al., 2018*).

## Whole-cell patch-clamp recording

Patch pipettes (4–6 MΩ) pulled from borosilicate glass were filled with 8 μl of RNAse-free internal solution containing in (mM): 144 K-gluconate, 3 $MgCl_2$, 0.5 EGTA, 10 HEPES, and pH 7.2 (285/295 mOsm).

Whole-cell recordings were performed at 25.3 ± 0.2°C using a patch-clamp amplifier (Axopatch 200B, Molecular Devices). Data were filtered at 5–10 kHz and digitized at 50 kHz using an acquisition board (Digidata 1440, Molecular Devices) attached to a personal computer running pCLAMP 10.2 software package (Molecular Devices). For ATP washout experiments neurons were recorded in voltage clamp mode using an ATP-free internal solution containing in (mM): 140 KCl, 20 NaCl, 2 $MgCl_2$, 10 EGTA, 10 HEPES, and pH 7.2.

## Cytoplasm harvesting and scRT-PCR

At the end of the whole-cell recording, lasting less than 15 min, the cytoplasmic content was aspirated in the recording pipette. The pipette's content was expelled into a test tube and reverse transcription (RT) was performed in a final volume of 10 µl, as described previously (*Lambolez et al., 1992*). The scRT-PCR protocol was designed to probe simultaneously the expression of neuronal markers, $K_{ATP}$ channels subunits or some key elements of lactate metabolism. Two-step amplification was performed essentially as described (*Cauli et al., 1997*; *Devienne et al., 2018*). Briefly, cDNAs present in the 10 µl reverse transcription reaction were first amplified simultaneously using all external primer pairs listed in the Key Ressources Table. Taq polymerase and 20 pmol of each primer were added to the buffer supplied by the manufacturer (final volume, 100 µl), and 20 cycles (94 °C, 30 s; 60 °C, 30 s; 72 °C, 35 s) of PCR were run. Second rounds of PCR were performed using 1 µl of the first PCR product as a template. In this second round, each cDNA was amplified individually using its specific nested primer pair (Key Ressources Table in Appendix 1) by performing 35 PCR cycles (as described above). 10 µl of each individual PCR product were run on a 2% agarose gel stained with ethidium bromide using ΦX174 digested by HaeIII as a molecular weight marker.

## Perforated patch-clamp recording

Gramicidin stock solution (2 mg/ml, Sigma-Aldrich) was prepared in DMSO and diluted to 10–20 µg/ml (*Zawar and Neumcke, 2000*) in the RNAse-free internal solution described above. The pipette tip was filled with gramicidin-free solution. Progress in perforation was evaluated by monitoring the capacitive transient currents elicited by −10 mV voltage pulses from a holding potential of −60 mV. In perforated patch configuration, a continuous current (52 ± 7 pA) was injected to induce the spiking of action potentials at stable firing rates of 4.1 ± 0.4 Hz obtained after an equilibration period of 3.6 ± 0.5 min. Membrane and access resistance were continuously monitored by applying −50 pA hyper-polarizing current pulses lasting 1 s every 10 s using an external stimulator (S900, Dagan) connected to the amplifier. Recordings were stopped when going into whole-cell configuration occurred, as evidenced by sudden increase of spike amplitude and decrease of access resistance.

## NADH imaging

Recordings were made in layers II–III of the rat somatosensory cortex. Widefield fluorescent images were obtained using a double port upright microscope BX51WI, WI-DPMC, Olympus with a ×60 objective (LUMPlan Fl/IR ×60 /0.90 W, Olympus) and a digital camera (CoolSnap HQ2, Roper Scientific) attached on the front port of the microscope. NADH autofluorescence was obtained by 365 nm excitation with a Light Emitting Device (LED, pE-2, CoolLED) using Imaging Workbench 6.0.25 software (INDEC Systems) and dichroic (FF395/495/610-Di01-25 × 36, Semrock) and emission filters (FF01-425/527/685-25, Semrock). Infrared Dodt gradient contrast images (IR-DGC, *Dodt and Ziegl-gänsberger, 1998*) were obtained using a 780 nm collimated LED (M780L3-C1,Thorlabs) as a trans-mitted light source and DGC optics (Luigs and Neumann). Autofluorescence and IR-DGC images were collected every 10 s by alternating the fluorescence and transmitted light sources. In parallel, infrared transmitted light images of slices were also continuously monitored on the back-port of the microscope using a customized beam splitter (725 DCSPXR, Semrock) and an analogic CCD camera (XC ST-70 CE, Sony). The focal plane was maintained constant online using infrared DGC images of cells as anatomical landmarks (*Lacroix et al., 2015*).

## Subcloning and viral production

The coding sequence of the ATP sensor ATeam1.03YEMK (*Imamura et al., 2009*) was subcloned into the viral vector pSinRep5. Sindbis virus was produced as previously described (*Piquet et al., 2018*). Recombinant pSinRep5 and helper plasmid pDH26S (Invitrogen) were transcribed in vitro into capped

RNA using the Megascript SP6 kit (Ambion). Baby hamster kidney-21 cells (BHK-21, clone 13, *Mesocricetus auratus*, hamster, Syrian golden), negative for mycoplasma contamination and purchased from ATCC (CCL-10, RRID:CVCL_1915, lot number 1545545), were only used for viral production. BHK-21 cells were electroporated with sensor-containing RNA and helper RNA ($2.10^7$ cells, 950 µF, 230 V) and incubated for 24 h at 37 °C in 5 % $CO_2$ in Dulbecco's modified Eagle medium supplemented with 5 % fetal calf serum before collecting cell supernatant containing the viruses. The virus titer ($10^8$ infectious particles/ml) was determined after counting fluorescent baby hamster kidney cells infected using serial dilution of the stock virus.

## Brain slice viral transduction

Brain slices were placed onto a millicell membrane (Millipore) with culture medium 50 % minimum essential medium, 50 % Hank's balanced salt sodium, 6.5 g/l glucose, and 100 U/ml penicillin–streptomycin (Sigma-Aldrich) as previously described (*Piquet et al., 2018*). Infection was performed by adding ~$5 \times 10^5$ particles per slice. Slices were incubated overnight at 35 °C in 5 % CO2. The next morning, brain slices were equilibrated for 1 hr in aCSF containing (in mM): 126 NaCl, 2.5 KCl, 1.25 $NaH_2PO_4$, 2 $CaCl_2$, 1 $MgCl_2$, 26 $NaHCO_3$, 10 glucose, and 15 sucrose. Slices were then placed into the recording chamber, heated at ~30 °C and continuously perfused at 1–2 ml/min.

## FRET imaging

Recordings were made from visually identified pyramidal cells in layers II–III of the rat somatosensory cortex. Widefield fluorescent images were obtained using a ×40 objective and a digital camera attached on the front port of the microscope. The ATP sensor ATeam1.03YEMK was excited at 400 nm with a LED using Imaging Workbench 6.0.25 software and excitation (FF02-438/24-25, Semrock) and dichroic filters (FF458-Di02-25 × 36, Semrock). Double fluorescence images were collected every 15 s by alternating the fluorescence emission filters for the CFP (FF01-483/32-25, Semrock) and the YFP (FF01-542/27-25, Semrock) using a filter wheel (Lambda 10B, Sutter Instruments). The focal plane was maintained constant online as described above.

## Pharmacological studies

Pinacidil (100 µM, Sigma-Aldrich); diazoxide (300 µM, Sigma-Aldrich) and tolbutamide (500 µM, Sigma-Aldrich), Mn(III)tetrakis(1-methyl-4-pyridyl)porphyrin (MnTMPyP, 25 µM, Millipore), 4-CIN (250 µM, Sigma-Aldrich); IAA (200 µM, Sigma-Aldrich) or KCN (1 mM, Sigma-Aldrich) was dissolved in aCSF from stock solutions of pinacidil (100 mM; NaOH 1 M), diazoxide (300 mM; NaOH 1 M), tolbutamide (500 mM; NaOH 1 M), 4-CIN (250 mM; DMSO), IAA (200 mM, water), and KCN (1 M, water). Changes in extracellular glucose, lactate, or pyruvate concentration were compensated by changes in sucrose concentration to maintain the osmolarity of the aCSF constant as previously described (*Miki et al., 2001*; *Miki et al., 2001*; arin et al., 2015; *Piquet et al., 2018*) and pH was adjusted to 7.4.

## Quantification and statistical analysis

### Analysis of somatic features

The laminar location determined by infrared videomicroscopy and recorded as 1–4 according to a location right within layers I, II/III, or IV. For neurons located at the border of layers I–II/III and II/III–IV, the laminar location was represented by 1.5 and 3.5, respectively. Somatic features were measured from IR DGC of the recorded neurons. Briefly, the soma was manually delineated using Image-Pro Analyzer 7.0 software (MediaCybernetics) and length of major and minor axes, perimeter and area were extracted. The soma elongation was calculated as the ratio between major and minor axis. Roundness was calculated according to: $\frac{perimeter^2}{4\pi \times area}$ ; a value close to one is indicative of round somata.

### Analysis of electrophysiological properties

Thirty-two electrophysiological properties chosen to describe the electrophysiological diversity of cortical neurons (*Petilla Interneuron Nomenclature Group et al., 2008*) were determined using the I-clamp fast mode of the amplifier as previously described (*Karagiannis et al., 2009*). Membrane potential values were corrected for theoretical liquid junction potential (−15.6 mV). Resting membrane potential was measured just after passing in whole-cell configuration, and only cells with a resting membrane potential more negative than −55 mV were analyzed further. Membrane resistance ($R_m$)

and membrane time constant ($\tau_m$) were determined on responses to hyperpolarizing current pulses (duration, 800 ms) eliciting voltage shifts of 10–15 mV negative to rest (*Kawaguchi, 1993*; *Kawaguchi, 1995*). Time constant was determined by fitting this voltage response to a single exponential. Membrane capacitance ($C_m$) was calculated according to $C_m = \tau_m/R_m$. Sag index was quantified as a relative decrease in membrane conductance according to $(G_{sag} - G_{hyp})/G_{sag}$ (*Halabisky et al., 2006*) where $G_{hyp}$ and $G_{sag}$ correspond to the whole-cell conductance when the sag was inactive and active, respectively. $G_{sag}$ was measured as the slope of the linear portion of a current–voltage (I–V) plot, where V was determined at the end of 800 ms hyperpolarizing current pulses (−100 to 0 pA) and $G_{hyp}$ as the slope of the linear portion of an I–V plot, where V was determined as the maximal negative potential during the 800 ms hyperpolarizing pulses. Rheobase was quantified as the minimal depolarizing current pulse intensity (800 ms duration pulses, 10 pA increments) generating at least one action potential. First spike latency (*Gupta et al., 2000*; *Petilla Interneuron Nomenclature Group et al., 2008*) was measured at rheobase as the time needed to elicit the first action potential. To describe different firing behaviors near threshold, spike frequency was measured near spike threshold on the first trace in which at least three spikes were triggered. Instantaneous discharge frequencies were measured and fitted to a straight line according to $F_{threshold} = m_{threshold}\ t + F_{min}$, where $m_{threshold}$ is the slope termed adaptation, $t$ the time, and $F_{min}$, the minimal steady-state frequency. Analysis of the action potentials waveforms was done on the first two spikes. Their amplitude (A1 and A2) was measured from threshold to the positive peak of the spike. Their duration (D1 and D2) was measured at half amplitude (*Kawaguchi, 1993*; *Cauli et al., 1997*). Their amplitude reduction and the duration increase were calculated according to $(A1 - A2)/A1$ and $(D2 - D1)/D1$, respectively (*Cauli et al., 1997*; *Cauli et al., 2000*). The amplitude and the latency of the fAH and mAH were measured for the first two action potentials as the difference between spike threshold and the negative peak of the AHs (*Kawaguchi, 1993*). The amplitude and latency of AD following single spikes (*Haj-Dahmane and Andrade, 1997*) were measured as the difference between the negative peak of the fAH and the peak of the AD and between the spike threshold and the peak of the AD, respectively. When neurons did not exhibit mAH or AD, amplitude and latency were arbitrarily set to 0. A complex spike amplitude accommodation during a train of action potentials, consisting in a transient decrease of spikes amplitude, was measured as the difference between the peak of the smallest action potential and the peak of the following largest action potential (*Cauli et al., 2000*). Maximal firing rate was defined as the last trace before prominent reduction of action potentials amplitude indicative of a saturated discharge. To take into account the biphasic spike frequency adaptation (early and late) occurring at high firing rates (*Cauli et al., 1997*; *Cauli et al., 2000*; *Gallopin et al., 2006*), instantaneous firing frequency was fitted to a single exponential (*Halabisky et al., 2006*) with a sloping baseline, according to: $F_{saturation}. = A_{sat}.e^{-t/\tau_{sat}} + t.m_{sat} + F_{max}$, where $A_{sat}$ corresponds to the amplitude of early frequency adaptation, $\tau_{sat}$ to the time constant of early adaptation, $m_{sat}$ to the slope of late adaptation, and $F_{max}$ to the maximal steady-state frequency.

## Unsupervised clustering

To classify neurons unsupervised clustering was performed using the laminar location of the soma, 10 molecular parameters (*Slc17a7*, *Gad2* and/or *Gad1*, *Nos1*, *Calb1*, *Pvalb*, *Calb2*, *Npy*, *Vip*, *Sst* and *Cck*) and the 32 electrophysiological parameters described above. Neurons positive for *Gad2* and/or *Gad1* were denoted as *Gad* positive and these mRNAs were considered as a single molecular variable as previously described (*Gallopin et al., 2006*). Parameters were standardized by centering and reducing all of the values. Cluster analysis was run on Statistica 6.1 software (Statsoft) using Ward's method (*Ward, 1963*). The final number of clusters was established by hierarchically subdividing the clustering tree into higher order clusters as previously described (*Karagiannis et al., 2009*).

## Analysis of voltage clamp recordings

Whole-cell currents were measured from a holding potential of −70 mV and membrane resistances were determined by applying a voltage step to −60 mV of 100 ms every 5 s. The effects of $K_{ATP}$ channel modulators were measured at the end of drug application by averaging, over a period of 1 min, whole-cell currents and changes in membrane resistance relative to control baseline prior to the application of drugs. Whole-cell $K_{ATP}$ current and conductance were determined by subtracting current and conductance measured under $K_{ATP}$ channel activator by their value measured under $K_{ATP}$ channel

blocker. The relative whole-cell $K_{ATP}$ conductance was determined by dividing the whole-cell $K_{ATP}$ conductance by the whole-cell conductance measured under $K_{ATP}$ channel activator. Whole-cell $K_{ATP}$ current density was determined by dividing the whole-cell $K_{ATP}$ current by the membrane capacitance. $K_{ATP}$ current reversal potential was measured by subtracting $I/V$ relationships obtained during voltage ramps from −60 to −130 mV determined under $K_{ATP}$ channel activator and blocker, respectively.

During ATP washout experiments, whole-cell currents and $I/V$ relationships were measured every 10 s at a holding potential of −50 mV and during voltage ramps from −40 to −120 mV, respectively. Washout currents were determined by subtracting the whole-cell currents measured at the beginning and the end of the whole-cell recording, respectively.

## Analysis of current clamp recordings

Every 10 s, membrane potential and mean firing rate were measured and membrane resistances were determined from voltage responses induced by –50 pA currents pulses lasting 1 s. $K_{ATP}$ voltage response and changes in membrane resistance and firing rate were determined by subtracting their value measured under $K_{ATP}$ channel activator by their value measured under $K_{ATP}$ channel blocker. Neurons were considered as responsive to $K_{ATP}$ channel modulators if the $K_{ATP}$ channel activator induced both a hyperpolarization and a decrease in membrane resistance reversed by the $K_{ATP}$ channel blocker.

## Analysis of perforated patch recordings

Mean firing frequency was measured every 10 s. Quantification of spiking activity was determined by averaging firing frequency over a period of 5 min preceding a change in extracellular aCSF composition. Firing frequencies were normalized by the averaged mean firing frequency measured under control condition.

## NADH imaging

Shading correction was applied offline on the NADH autofluorescence images using the 'Shading Corrector' plugin of FIJI software (*Schindelin et al., 2012*) and a blank field reference image. To compensate for potential $x$–$y$ drifts all IR-DGC images were realigned offline using the 'StackReg' and 'TurboReg' plugins (*Thévenaz et al., 1998*) of FIJI software and the same registration was applied to the corrected NADH autofluorescence images. To determine somatic regions of interest (ROIs) the soma was manually delineated on IR-DGC images. The mean NADH autofluorescence was measured at each time point using the same ROIs. Variations of fluorescence intensity were expressed as the ratio $(F − F_0)/F_0$ where $F$ corresponds to the mean fluorescence intensity in the ROI at a given time point, and F0 corresponds to the mean fluorescence intensity in the same ROI during the 5 min control baseline prior to changes in aCSF composition. Effect of monocarboxylate superfusion or oxidative phosphorylation blockade was quantified by averaging the normalized ratio $(R/R_0)$ during the last 5 min of drug application.

## FRET imaging

All images were realigned offline as described above using the YFP images as the reference for registration. Fluorescence ratios were calculated by dividing the registered YFP images by the registered CFP images using FIJI. The somatic ROIs were manually delineated on the YFP images as described above. The mean ratio was measured at each time point using the same ROIs. Variations of fluorescence ratio were expressed as the ratio $(R − R_0)/R_0$ where $R$ corresponds to the fluorescence ratio in the ROI at a given time point, and $R_0$ corresponds to the mean fluorescence ratio in the same ROI during the 10 min control baseline prior to drug application. Effect of glycolysis or oxidative phosphorylation blockade was quantified by averaging the normalized ratio during the last 5 min of drug application.

## Statistical analysis

Statistical analyses were performed with Statistica 6.1 and GraphPad Prism 7. All values are expressed as means ± SEM. Normality of distributions and equality of variances were assessed using the Shapiro–Wilk test and the Fisher $F$-test, respectively. Parametric tests were only used if these criteria were met. Holm–Bonferroni correction was used for multiple comparisons and p values are given as uncorrected.

Statistical significance on all figures uses the following convention of corrected p values: $*p < 0.05$, $**p < 0.01$, $***p < 0.001$.

Statistical significance of morphological and electrophysiological properties of neurons was determined using the Mann–Whitney $U$ test. Comparison of the occurrence of expressed genes and of responsiveness of $K_{ATP}$ channel modulators between different cell types was determined using Fisher's exact test. Statistical significance of the effects of $K_{ATP}$ channel modulators was determined using the Friedman and post hoc Dunn's tests. Significance of the effect of the ROS scavenger was determined using one-tailed unpaired Student's $t$-test. Comparison of $K_{ATP}$ channel properties was determined using Mann–Whitney $U$, Student's $t$, or Kruskal–Wallis $H$ tests. Comparison of responses between $Kcnj11^{+/+}$ and $Kcnj11^{-/-}$ neurons was determined using Mann–Whitney $U$ test. Statistical significance of the effects of energy substrates and drug applications on evoked firing in perforated patch recordings was determined using Friedman and Dunn's tests. Comparison of the effects of monocarboxylates and cyanide on NADH fluorescence was determined using Mann–Whitney $U$ test. Statistical significance of the effects of metabolic inhibitors on intracellular ATP was determined using Friedman and Dunn's tests.

## Acknowledgements

This work was supported by grants from the Human Frontier Science Program (HFSP, RGY0070/2007, BC) and the Agence Nationale pour la Recherche (ANR 2011 MALZ 003 01, BC). AK was supported by a Fondation pour la Recherche Médicale fellowship (FDT20100920106). BLG was supported by a Fondation pour la Recherche sur Alzheimer fellowship. We thank the animal facility of the IBPS (Paris, France).

## Additional information

### Funding

| Funder | Grant reference number | Author |
| --- | --- | --- |
| Human Frontier Science Program | RGY0070/2007 | Bruno Cauli |
| Agence Nationale de la Recherche | ANR 2011 MALZ 003 01 | Bruno Cauli |
| Fondation pour la Recherche Médicale | FDT20100920106 | Anastassios Karagiannis |
| Fondation pour la Recherche sur Alzheimer | | Benjamin Le Gac |

The funders had no role in study design, data collection and interpretation, or the decision to submit the work for publication.

### Author contributions

Anastassios Karagiannis, Investigation, Visualization; Thierry Gallopin, Conceptualization, Investigation, Writing - review and editing; Alexandre Lacroix, Fabrice Plaisier, Juliette Piquet, Hélène Geoffroy, Régine Hepp, Investigation; Jérémie Naudé, Formal analysis, Writing - review and editing; Benjamin Le Gac, Richard Egger, Dongdong Li, Jochen F Staiger, Writing - review and editing; Bertrand Lambolez, Jean Rossier, Resources, Writing - review and editing; Hiromi Imamura, Susumu Seino, Resources; Jochen Roeper, Conceptualization, Resources, Writing - review and editing; Bruno Cauli, Conceptualization, Formal analysis, Funding acquisition, Investigation, Project administration, Resources, Supervision, Visualization, Writing - original draft, Writing - review and editing

### Author ORCIDs

Jérémie Naudé http://orcid.org/0000-0001-5781-6498
Bertrand Lambolez http://orcid.org/0000-0002-0653-480X
Jean Rossier http://orcid.org/0000-0003-1821-2135
Hiromi Imamura http://orcid.org/0000-0002-1896-0443

Jochen Roeper (iD) http://orcid.org/0000-0003-2145-8742
Bruno Cauli (iD) http://orcid.org/0000-0003-1471-4621

### Ethics

Wistar rats, C57BL/6RJ or Kcnj11-/- (B6.129P2-Kcnj11tm1Sse, backcrossed into C57BL6 over six generations) mice were used for all experiments in accordance with French regulations (Code Rural R214/87 to R214/130) and conformed to the ethical guidelines of both the directive 2010/63/EU of the European Parliament and of the Council and the French National Charter on the ethics of animal experimentation. A maximum of three rats or five mice were housed per cage and single animal housing was avoided. Male rats and mice of both genders were housed on a 12-hr light/dark cycle in a temperature-controlled (21–25°C) room and were given food and water ad libitum. Animals were used for experimentation at 13–24 days of age.

### Decision letter and Author response

Decision letter https://doi.org/10.7554/eLife.71424.sa1
Author response https://doi.org/10.7554/eLife.71424.sa2

---

# Additional files

### Supplementary files

- Supplementary file 1. Somatic properties of different neuronal types $n$, number of cells, < significantly smaller with $p \leq 0.05$; << significantly smaller with $p \leq 0.01$; <<< significantly smaller with $p \leq 0.001$. n.s., not statistically significant.

- Supplementary file 2. Detection rate of molecular markers in different neuronal types. Detection rates are given in %; $n$, number of cells; > significantly larger with $p \leq 0.05$; >> significantly larger with $p \leq 0.01$; >>> significantly larger with $p \leq 0.001$. n.s., not statistically significant.

- Supplementary file 3. Passive properties of different neuronal types $n$, number of cells, < significantly smaller with $p \leq 0.05$; << significantly smaller with $p \leq 0.01$; <<< significantly smaller with $p \leq 0.001$.

- Supplementary file 4. Just above threshold properties of different neuronal types $n$, number of cells; < significantly smaller with $p \leq 0.05$; << significantly smaller with $p \leq 0.01$; <<< significantly smaller with $p \leq 0.001$.

- Supplementary file 5. Firing properties of different neuronal types $n$, number of cells; < significantly smaller with $p \leq 0.05$; << significantly smaller with $p \leq 0.01$; <<< significantly smaller with $p \leq 0.001$.

- Supplementary file 6. Action potentials properties of different neuronal types $n$, number of cells; < significantly smaller with $p \leq 0.05$; << significantly smaller with $p \leq 0.01$; <<< significantly smaller with $p \leq 0.001$.

- Supplementary file 7. AH and AD properties of different neuronal types $n$, number of cells; < significantly smaller with $p \leq 0.05$; << significantly smaller with $p \leq 0.01$; <<< significantly smaller with $p \leq 0.001$.

- Transparent reporting form

- Source data 1. Statistcal comparisons of somatic properties in different neuronal types.

- Source data 2. Statistcal comparisons of detection rate of molecular markers in different neuronal types.

- Source data 3. Statistcal comparisons of passive properties in different neuronal types.

- Source data 4. Statistcal comparisons of just above threshold properties in different neuronal types.

- Source data 5. Statistcal comparisons of firing properties in different neuronal types.

- Source data 6. Statistcal comparisons of action potentials properties in different neuronal types.

- Source data 7. Statistcal comparisons of AH and AD properties in different neuronal types.

## Data availability

All data generated or analysed during this study are included in the manuscript and supporting files. Source data files have been provided for Figures 1 to 6.

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

# Appendix 1

## Appendix 1—key resources table

| Reagent type (species) or resource | Designation | Source or reference | Identifiers | Additional information |
|---|---|---|---|---|
| Strain, strain background (*Rattus norvegicus,* Wistar, male) | Wistar | Janvier Labs | jHan:WI | |
| Strain, strain background (*Mus musculus*, C57BL/6RJ, male and female) | Wild type, *Kcnj11* [+/+] | Janvier Labs | C57BL/6RJ | |
| Strain, strain background (*Mus musculus*, B6.129P2, male and female) | B6.129P2-*Kcnj11*[tm1Sse], Kcnj11[−/−] | PMID: 9724715 (*Miki et al., 1998*) | RRID: MGI:5433111 | |
| Cell line (*Mesocricetus auratus*) | BHK-21 clone 13 (baby hamster kidneys fibroblasts) | ATCC | CCL-10, RRID: CVCL_1915 | |
| Recombinant DNA reagent | pcDNA-ATeam1.03YEMK (plasmid) | PMID:19720993( *Imamura et al., 2009*) | | |
| Recombinant DNA reagent | pSinRep5 (plasmid) | Invitrogen | K750-01 | |
| Recombinant DNA reagent | pDH(26 S) (helper plasmid) | Invitrogen | K750-01 | |
| Sequence-based reagent | rat *Slc17a7* external sense | PMID:16339088 (*Gallopin et al., 2006*) | PCR primers | GGCTCCTTTTTCTGGGGGTAC |
| Sequence-based reagent | rat *Slc17a7* external antisense | PMID:16339088 (*Gallopin et al., 2006*) | PCR primers | CCAGCCGACTCCGTTCTAAG |
| Sequence-based reagent | rat *Slc17a7* internal sense | PMID:16339088 (*Gallopin et al., 2006*) | PCR primers | TGGGGGTACATTGTCACTCAGA |
| Sequence-based reagent | rat *Slc17a7* internal antisense | PMID:16339088 (*Gallopin et al., 2006*) | PCR primers | ATGGCAAGCAGGGTATGTGAC |
| Sequence-based reagent | rat/mouse *Gad2* external sense | PMID:19295167 (*Karagiannis et al., 2009*) | PCR primers | CCAAAAGTTCACGGGCGG |
| Sequence-based reagent | rat/mouse *Gad2* external antisense | PMID:19295167 (*Karagiannis et al., 2009*) | PCR primers | TCCTCCAGATTTTGCGGTTG |
| Sequence-based reagent | rat *Gad2* internal sense | PMID:19295167 (*Karagiannis et al., 2009*) | PCR primers | TGAGAAGCCAGCAGAGAGCG |
| Sequence-based reagent | rat *Gad2* internal antisense | PMID:19295167 (*Karagiannis et al., 2009*) | PCR primers | TGGGGTAATGGAAATCAATCACTT |
| Sequence-based reagent | rat *Gad1* external sense | PMID:19295167 (*Karagiannis et al., 2009*) | PCR primers | ATGATACTTGGTGTGGCGTAGC |
| Sequence-based reagent | rat *Gad1* external antisense | PMID:19295167 (*Karagiannis et al., 2009*) | PCR primers | GTTTGCTCCTCCCCGTTCTTAG |
| Sequence-based reagent | rat *Gad1* internal sense | PMID:19295167 (*Karagiannis et al., 2009*) | PCR primers | CAATAGCCTGGAAGAGAAGAGTCG |
| Sequence-based reagent | rat *Gad1* internal antisense | PMID:19295167 (*Karagiannis et al., 2009*) | PCR primers | GTTTGCTCCTCCCCGTTCTTAG |
| Sequence-based reagent | rat *Nos1* external sense | PMID:19295167 (*Karagiannis et al., 2009*) | PCR primers | CCTGGGGCTCAAATGGTATG |
| Sequence-based reagent | rat *Nos1* external antisense | PMID:19295167 (*Karagiannis et al., 2009*) | PCR primers | CACAATCCACACCCAGTCGG |
| Sequence-based reagent | rat *Nos1* internal sense | PMID:19295167 (*Karagiannis et al., 2009*) | PCR primers | CCTCCCCGCTGTGTCCAA |
| Sequence-based reagent | rat *Nos1* internal antisense | PMID:19295167 (*Karagiannis et al., 2009*) | PCR primers | GAGTGGTGGTCAACGATGGTCA |
| Sequence-based reagent | rat *Calb1* external sense | PMID:19295167 (*Karagiannis et al., 2009*) | PCR primers | GAAAGAAGGCTGGATTGGAG |

*Appendix 1 Continued on next page*

*Appendix 1 Continued*

| Reagent type (species) or resource | Designation | Source or reference | Identifiers | Additional information |
|---|---|---|---|---|
| Sequence-based reagent | rat *Calb1* external antisense | PMID:19295167 (**Karagiannis et al., 2009**) | PCR primers | CCCACACATTTTGATTCCCTG |
| Sequence-based reagent | rat *Calb1* internal sense | PMID:19295167 (**Karagiannis et al., 2009**) | PCR primers | ATGGGCAGAGAGATGATGGG |
| Sequence-based reagent | rat *Calb1* internal antisense | PMID:19295167 (**Karagiannis et al., 2009**) | PCR primers | TATCATCCACGGTCTTGTTTGC |
| Sequence-based reagent | rat *Pvalb* external sense | PMID:19295167 (**Karagiannis et al., 2009**) | PCR primers | GCCTGAAGAAAAAGAGTGCGG |
| Sequence-based reagent | rat *Pvalb* external antisense | PMID:19295167 (**Karagiannis et al., 2009**) | PCR primers | GTCCCCGTCCTTGTCTCCAG |
| Sequence-based reagent | rat *Pvalb* internal sense | PMID:19295167 (**Karagiannis et al., 2009**) | PCR primers | GCGGATGATGTGAAGAAGGTG |
| Sequence-based reagent | rat *Pvalb* internal antisense | PMID:19295167 (**Karagiannis et al., 2009**) | PCR primers | CAGCCATCAGCGTCTTTGTT |
| Sequence-based reagent | rat *Calb2* external sense | PMID:19295167 (**Karagiannis et al., 2009**) | PCR primers | TTGATGCTGACGGAAATGGGTA |
| Sequence-based reagent | rat *Calb2* external antisense | PMID:19295167 (**Karagiannis et al., 2009**) | PCR primers | CAAGCCTCCATAAACTCAGCG |
| Sequence-based reagent | rat *Calb2* internal sense | PMID:19295167 (**Karagiannis et al., 2009**) | PCR primers | GCTGGAGAAGGCAAGGAAAGG |
| Sequence-based reagent | rat *Calb2* internal antisense | PMID:19295167 (**Karagiannis et al., 2009**) | PCR primers | ATTCTCTTCGGTTGGCAGGA |
| Sequence-based reagent | rat *Npy* external sense | PMID:19295167 (**Karagiannis et al., 2009**) | PCR primers | CGAATGGGGCTGTGTGGA |
| Sequence-based reagent | rat *Npy* external antisense | PMID:19295167 (**Karagiannis et al., 2009**) | PCR primers | AGTTTCATTTCCCATCACCACAT |
| Sequence-based reagent | rat *Npy* internal sense | PMID:19295167 (**Karagiannis et al., 2009**) | PCR primers | CCCTCGCTCTATCCCTGCTC |
| Sequence-based reagent | rat *Npy* internal antisense | PMID:19295167 (**Karagiannis et al., 2009**) | PCR primers | GTTCTGGGGGCATTTTCTGTG |
| Sequence-based reagent | rat *Vip* external sense | PMID:19295167 (**Karagiannis et al., 2009**) | PCR primers | TTATGATGTGTCCAGAAATGCGAG |
| Sequence-based reagent | rat *Vip* external antisense | PMID:19295167 (**Karagiannis et al., 2009**) | PCR primers | TTTTATTTGGTTTTGCTATGGAAG |
| Sequence-based reagent | rat *Vip* internal sense | PMID:19295167 (**Karagiannis et al., 2009**) | PCR primers | TGGCAAACGAATCAGCAGTAGC |
| Sequence-based reagent | rat *Vip* internal antisense | PMID:19295167 (**Karagiannis et al., 2009**) | PCR primers | GAATCTCCCTCACTGCTCCTCT |
| Sequence-based reagent | rat *Sst* external sense | PMID:19295167 (**Karagiannis et al., 2009**) | PCR primers | ATGCTGTCCTGCCGTCTCCA |
| Sequence-based reagent | rat *Sst* external antisense | PMID:17068095 (**Férézou et al., 2007**) | PCR primers | GCCTCATCTCGTCCTGCTCA |
| Sequence-based reagent | rat *Sst* internal sense | PMID:19295167 (**Karagiannis et al., 2009**) | PCR primers | GCATCGTCCTGGCTTTGGG |
| Sequence-based reagent | rat *Sst* internal antisense | PMID:19295167 (**Karagiannis et al., 2009**) | PCR primers | AGGCTCCAGGGCATCGTTCT |
| Sequence-based reagent | rat *Cck* external sense | PMID:19295167 (**Karagiannis et al., 2009**) | PCR primers | TGTCTGTGCGTGGTGATGGC |
| Sequence-based reagent | rat *Cck* external antisense | PMID:19295167 (**Karagiannis et al., 2009**) | PCR primers | GCATAGCAACATTAGGTCTGGGAG |
| Sequence-based reagent | rat *Cck* internal sense | PMID:19295167 (**Karagiannis et al., 2009**) | PCR primers | ATACATCCAGCAGGTCCGCAA |
| Sequence-based reagent | rat *Cck* internal antisense | PMID:19295167 (**Karagiannis et al., 2009**) | PCR primers | GGTCGTGTGCGTGGTTGTTT |
| Sequence-based reagent | rat *Kcnj8* external sense | This paper | PCR primers | CTGGCTCACAAGAACATCCG |

*Appendix 1 Continued on next page*

*Appendix 1 Continued*

| Reagent type (species) or resource | Designation | Source or reference | Identifiers | Additional information |
|---|---|---|---|---|
| Sequence-based reagent | rat *Kcnj8* external antisense | This paper | PCR primers | AGCGTCTCTGCCCTTCTGTG |
| Sequence-based reagent | rat *Kcnj8* internal sense | PMID:26156991 (*Varin et al., 2015*) | PCR primers | GCTGGCTGCTCTTCGCTATC |
| Sequence-based reagent | rat *Kcnj8* internal antisense | This paper | PCR primers | TTCTCCCTCCAAACCCAATG |
| Sequence-based reagent | rat *Kcnj11* external sense | This paper | PCR primers | CCCCACACGCTGCTCATTTT |
| Sequence-based reagent | rat *Kcnj11* external antisense | This paper | PCR primers | AGGAGCCAGGTCGTAGAGCG |
| Sequence-based reagent | rat *Kcnj11* internal sense | This paper | PCR primers | GCGTCACAAGCATCCACTCC |
| Sequence-based reagent | rat *Kcnj11* internal antisense | This paper | PCR primers | CCACCCACACCGTTCTCCAT |
| Sequence-based reagent | rat *Abcc8* external sense | This paper | PCR primers | GGTGAAGAAGCCTCCGATGA |
| Sequence-based reagent | rat *Abcc8* external antisense | This paper | PCR primers | GGTGAAGAAGCCTCCGATGA |
| Sequence-based reagent | rat *Abcc8* internal sense | This paper | PCR primers | GGTTCGGTCCACTGTCAAGG |
| Sequence-based reagent | rat *Abcc8* internal antisense | This paper | PCR primers | GTCAGCGTCTCCATCCGTGC |
| Sequence-based reagent | rat *Abcc9* external sense | This paper | PCR primers | CGCTGCCTTTTGAGTCCTGT |
| Sequence-based reagent | rat *Abcc9* external antisense | This paper | PCR primers | GATGGCAAGGAGGAGAGACG |
| Sequence-based reagent | rat *Abcc9* internal sense | This paper | PCR primers | TGGACAACTACGAGCAGGCG |
| Sequence-based reagent | rat *Abcc9* internal antisense | This paper | PCR primers | CACAACCCACCTGACCCACA |
| Sequence-based reagent | rat *Sst* intron external sense | PMID:17267760 (*Hill et al., 2007*) | PCR primers | GGAAATGGCTGGGACTCGTC |
| Sequence-based reagent | rat *Sst* intron external antisense | PMID17267760 (*Hill et al., 2007*) | PCR primers | AAACCATGGATGATAGGAAGTCGT |
| Sequence-based reagent | rat *Sst* intron internal sense | This paper | PCR primers | GTCCCCTTTGCGAATTCCCT |
| Sequence-based reagent | rat *Sst* intron internal antisense | This paper | PCR primers | TTCGAGCAGCTCCATTTTCC |
| Sequence-based reagent | rat SUR2A/B sense | This paper | PCR primers | ACTTCAGCGTTGGACAGAGACA |
| Sequence-based reagent | rat SUR2A/B antisense | This paper | PCR primers | GGTCAGCAGTCAGAATGGTGTG |
| Sequence-based reagent | mouse *Slc17a7* external sense | PMID:23565079 (*Cabezas et al., 2013*) | PCR primers | GGCTCCTTTTTCTGGGGCTAC |
| Sequence-based reagent | mouse *Slc17a7* external antisense | PMID:23565079 (*Cabezas et al., 2013*) | PCR primers | CCAGCCGACTCCGTTCTAAG |
| Sequence-based reagent | mouse *Slc17a7* internal sense | PMID:23565079 (*Cabezas et al., 2013*) | PCR primers | ATTCGCAGCCAACAGGGTCT |
| Sequence-based reagent | mouse *Slc17a7* internal antisense | PMID:23565079 (*Cabezas et al., 2013*) | PCR primers | TGGCAAGCAGGGTATGTGAC |
| Sequence-based reagent | mouse *Gad2* external sense | PMID:22754499 (*Perrenoud et al., 2012*) | PCR primers | CCAAAAGTTCACGGGCGG |
| Sequence-based reagent | mouse *Gad2* external antisense | PMID:22754499 (*Perrenoud et al., 2012*) | PCR primers | TCCTCCAGATTTTGCGGTTG |
| Sequence-based reagent | mouse *Gad2* internal sense | PMID:22754499 (*Perrenoud et al., 2012*) | PCR primers | CACCTGCGACCAAAAACCCT |

*Appendix 1 Continued on next page*

*Appendix 1 Continued*

| Reagent type (species) or resource | Designation | Source or reference | Identifiers | Additional information |
|---|---|---|---|---|
| Sequence-based reagent | mouse *Gad2* internal antisense | PMID:22754499 (*Perrenoud et al., 2012*) | PCR primers | GATTTTGCGGTTGGTCTGCC |
| Sequence-based reagent | mouse *Gad1* external sense | PMID:12196560 (*Férézou et al., 2002*) | PCR primers | TACGGGGTTCGCACAGGTC |
| Sequence-based reagent | mouse *Gad1* external antisense | PMID:23565079 (*Cabezas et al., 2013*) | PCR primers | CCCAGGCAGCATCCACAT |
| Sequence-based reagent | mouse *Gad1* internal sense | PMID:23565079 (*Cabezas et al., 2013*) | PCR primers | CCCAGAAGTGAAGACAAAAGGC |
| Sequence-based reagent | mouse *Gad1* internal antisense | PMID:23565079 (*Cabezas et al., 2013*) | PCR primers | AATGCTCCGTAAACAGTCGTGC |
| Sequence-based reagent | mouse *Atp1a1* external sense | PMID:29985318 (*Devienne et al., 2018*) | PCR primers | CAGGGCAGTGTTTCAGGCTAAS |
| Sequence-based reagent | mouse *Atp1a1* external antisense | PMID:29985318(*Devienne et al., 2018*) | PCR primers | CCGTGGAGAAGGATGGAGC |
| Sequence-based reagent | mouse *Atp1a1* internal sense | PMID:29985318 (*Devienne et al., 2018*) | PCR primers | TAAGCGGGCAGTAGCGGG |
| Sequence-based reagent | mouse *Atp1a1* internal antisense | PMID:29985318 (*Devienne et al., 2018*) | PCR primers | AGGTGTTTGGGCTCAGATGC |
| Sequence-based reagent | mouse *Atp1a2* external sense | PMID:29985318 (*Devienne et al., 2018*) | PCR primers | AGTGAGGAAGATGAGGGACAGG |
| Sequence-based reagent | mouse *Atp1a2* external antisense | PMID:29985318 (*Devienne et al., 2018*) | PCR primers | ACAGAAGCCCAGCACTCGTT |
| Sequence-based reagent | mouse *Atp1a2* internal sense | PMID:29985318 (*Devienne et al., 2018*) | PCR primers | AAATCCCCTTCAACTCCACCA |
| Sequence-based reagent | mouse *Atp1a2* internal antisense | PMID:29985318 (*Devienne et al., 2018*) | PCR primers | GTTCCCCAAGTCCTCCCAGC |
| Sequence-based reagent | mouse *Atp1a3* external sense | PMID:29985318 (*Devienne et al., 2018*) | PCR primers | CGGAAATACAATACTGACTGCGTG |
| Sequence-based reagent | mouse *Atp1a3* external antisense | PMID:29985318 (*Devienne et al., 2018*) | PCR primers | GTCATCCTCCGTCCCTGCC |
| Sequence-based reagent | mouse *Atp1a3* internal sense | PMID:29985318 (*Devienne et al., 2018*) | PCR primers | TGACACACAGTAAAGCCCAGGA |
| Sequence-based reagent | mouse *Atp1a3* internal antisense | PMID:29985318 (*Devienne et al., 2018*) | PCR primers | CCACAGCAGGATAGAGAAGCCA |
| Sequence-based reagent | mouse *Kcnj11* external sense | PMID:29985318 (*Devienne et al., 2018*) | PCR primers | CGGAGAGGGCACCAATGT |
| Sequence-based reagent | mouse *Kcnj11* external antisense | PMID:29985318 (*Devienne et al., 2018*) | PCR primers | CACCCACGCCATTCTCCA |
| Sequence-based reagent | mouse *Kcnj11* internal sense | PMID:29985318 (*Devienne et al., 2018*) | PCR primers | CATCCACTCCTTTTCATCTGCC |
| Sequence-based reagent | mouse *Kcnj11* internal antisense | PMID:29985318 (*Devienne et al., 2018*) | PCR primers | TCGGGGCTGGTGGTCTTG |
| Sequence-based reagent | mouse *Abcc8* external sense | PMID:29985318 (*Devienne et al., 2018*) | PCR primers | CAGTGTGCCCCCCGAGAG |
| Sequence-based reagent | mouse *Abcc8* external antisense | PMID:29985318 (*Devienne et al., 2018*) | PCR primers | GGTCTTCTCCCTCGCTGTCTG |
| Sequence-based reagent | mouse *Abcc8* internal sense | PMID:29985318 (*Devienne et al., 2018*) | PCR primers | ATCATCGGAGGCTTCTTCACC |
| Sequence-based reagent | mouse *Abcc8* internal antisense | PMID:29985318 (*Devienne et al., 2018*) | PCR primers | GGTCTTCTCCCTCGCTGTCTG |
| Sequence-based reagent | mouse *Sst* intron external sense | PMID:12930808 (*Thoby-Brisson et al., 2003*) | PCR primers | CTGTCCCCCTTACGAATCCC |
| Sequence-based reagent | mouse *Sst* intron external antisense | PMID:12930808 (*Thoby-Brisson et al., 2003*) | PCR primers | CCAGCACCAGGGATAGAGCC |
| Sequence-based reagent | mouse *Sst* intron internal sense | PMID:20427660 *Cea-del Rio et al., 2010* | PCR primers | CTTACGAATCCCCCAGCCTT |

*Appendix 1 Continued on next page*

*Appendix 1 Continued*

| Reagent type (species) or resource | Designation | Source or reference | Identifiers | Additional information |
|---|---|---|---|---|
| Sequence-based reagent | mouse *Sst* intron internal antisense | PMID:20427660 (***Cea-del Rio et al., 2010***) | PCR primers | TTGAAAGCCAGGGAGGAACT |
| Sequence-based reagent | rat *Slc16a1* external sense | This paper | PCR primers | GTCAGCCTTCCTCCTTTCCA |
| Sequence-based reagent | rat *Slc16a1* external antisense | This paper | PCR primers | TCCGCTTTCTGTTCTTTGGC |
| Sequence-based reagent | rat *Slc16a1* internal sense | This paper | PCR primers | TTGTTGCGAATGGAGTGTGC |
| Sequence-based reagent | rat *Slc16a1* internal antisense | This paper | PCR primers | CACGCCACAAGCCCAGTATG |
| Sequence-based reagent | rat *Slc16a7* external sense | This paper | PCR primers | GCGAAGTCTAAAAGTAAGGTTGGC |
| Sequence-based reagent | rat *Slc16a7* external antisense | This paper | PCR primers | ATTTACCAGCCAGGGGAGGG |
| Sequence-based reagent | rat *Slc16a7* internal sense | This paper | PCR primers | CCGTATGCTAAGGACAAAGGAGT |
| Sequence-based reagent | rat *Slc16a7* internal antisense | This paper | PCR primers | GGGAAGAACTGGGCAACACT |
| Sequence-based reagent | rat *Slc16a3* external sense | This paper | PCR primers | CATTGGTCTCGTGCTGCTGTS |
| Sequence-based reagent | rat *Slc16a3* external antisense | This paper | PCR primers | CCCCGTTTTTCTCAGGCTCT |
| Sequence-based reagent | rat *Slc16a3* internal sense | This paper | PCR primers | TGTGGCTGTGCTCATCGGAC |
| Sequence-based reagent | rat *Slc16a3* internal antisense | This paper | PCR primers | CCTCTTCCTCTTCCCGATGC |
| Sequence-based reagent | rat *Ldha* external sense | This paper | PCR primers | GAAGAACAGGTCCCCCAGAA |
| Sequence-based reagent | rat *Ldha* external antisense | This paper | PCR primers | GGGTTTGAGACGATGAGCAGT |
| Sequence-based reagent | rat *Ldha* internal sense | This paper | PCR primers | CAGTTGTTGGGGTTGGTGCT |
| Sequence-based reagent | rat *Ldha* internal antisense | This paper | PCR primers | TCTCTCCCTCTTGCTGACGG |
| Sequence-based reagent | rat *Ldhb* external sense | This paper | PCR primers | ACTGCCGTCCCGAACAACAA |
| Sequence-based reagent | rat *Ldhb* external antisense | This paper | PCR primers | ACTCTCCCCCTCCTGCTGG |
| Sequence-based reagent | rat *Ldhb* internal sense | This paper | PCR primers | TCTGGGGAAGTCTCTGGCTGA |
| Sequence-based reagent | rat *Ldhb* internal antisense | This paper | PCR primers | TTGGCTGTCACGGAGTAATCTTT |
| Commercial assay or kit | MEGAscript SP6 Transcription Kit | Ambion | AM1330 | |
| Chemical compound, drug | Pinacidil monohydrate | Sigma-Aldrich | P154 | |
| Chemical compound, drug | Diazoxide | Sigma-Aldrich | D9035 | |
| Chemical compound, drug | Tolbutamide | Sigma-Aldrich | T0891 | |
| Chemical compound, drug | Mn(III)tetrakis(1-methyl-4-pyridyl) porphyrin | Millipore | 475,872 | |
| Chemical compound, drug | Gramicidin from Bacillus aneurinolyticus (Bacillus brevis) | Sigma-Aldrich | G5002 | |

*Appendix 1 Continued on next page*

*Appendix 1 Continued*

| Reagent type (species) or resource | Designation | Source or reference | Identifiers | Additional information |
|---|---|---|---|---|
| Chemical compound, drug | Sodium L-lactate | Sigma-Aldrich | L7022 | |
| Chemical compound, drug | $\alpha$-Cyano-4-hydroxycinnamic Acid | Sigma-Aldrich | C2020 | |
| Chemical compound, drug | Sodium pyruvate | Sigma-Aldrich | P2256 | |
| Chemical compound, drug | Sodium iodoacetate | Sigma-Aldrich | I2512 | |
| Chemical compound, drug | Potassium cyanide | Sigma-Aldrich | 60,178 | |
| Chemical compound, drug | Dithiothreitol | VWR | 443,852 A | |
| Chemical compound, drug | Primer 'random' | | 4731001 | |
| Chemical compound, drug | dNTPs | GE Healthcare Life Sciences | 28-4065-52 | |
| Chemical compound, drug | Mineral Oil | Sigma-Aldrich | M5904 | |
| Chemical compound, drug | RNasin Ribonuclease Inhibitors | Promega | N2511 | |
| Chemical compound, drug | SuperScript II Reverse Transcriptase | Invitrogen | 18064014 | |
| Chemical compound, drug | Taq DNA Polymerase | Qiagen | 201,205 | |
| Chemical compound, drug | Penicillin-Streptomycin | Sigma-Aldrich | P4333-100ML | |
| Software, algorithm | Pclamp v 10.2 | Molecular Devices | RRID: SCR_011323 | |
| Software, algorithm | Matlab v 2018b | MathWorks | RRID: SCR_001622 | |
| Software, algorithm | Statistica v 6.1 | Statsoft | RRID: SCR_014213 | |
| Software, algorithm | GraphPad Prism v 7 | GraphPad | RRID: SCR_002798 | |
| Software, algorithm | ImagingWorkbench v 6.0.25 | INDEC Systems | | |
| Software, algorithm | FIJI | PMID:22743772 (**Schindelin et al., 2012**) | RRID: SCR_002285 | MID:29985318 |
| Software, algorithm | Image-Pro Analyzer v 7 | MediaCybernetics | | |
| Other | Vibratome | Leica | VT1000S RRID: SCR_016495 | |
| Other | Upright microscope | Olympus | BX51WI | |
| Other | Dual port module | Olympus | WI-DPMC | |
| Other | ×60 Objective | Olympus | LUMPlan Fl /IR 60 x/0.90 W | |
| Other | ×40 Objetive | Olympus | LUMPlan Fl /IR 40 x/0.80 W | |
| Other | CCD camera | Roper Scientific | CoolSnap HQ2 | |
| Other | Axopatch 200B | Molecular Devices | RRID: SCR_018866 | |
| Other | Digidata 1440A | Molecular Devices | RRID: SCR_021038 | |
| Other | S900 stimulator | Dagan Corporation | | |

*Appendix 1 Continued on next page*

*Appendix 1 Continued*

| Reagent type (species) or resource | Designation | Source or reference | Identifiers | Additional information |
|---|---|---|---|---|
| Other | pE-2 | CoolLED | | |
| Other | Dichroic mirror | Semrock | FF395/495/610-Di01-25 × 36 | |
| Other | Emission filter | Semrock | FF01-425/527/685-25 | |
| Other | 780 nm Collimated LED | Thorlabs | M780L3-C1 | |
| Other | Dodt Gradient Contrast | Luigs and Neumann | 200-100 200 0155 | |
| Other | Beam splitter | Semrock | 725 DCSPXR | |
| Other | Analogic CCD camera | Sony | XC ST-70 CE | |
| Other | Millicell | Millipore | PICM0RG50 | |
| Other | Excitation filter | Semrock | FF02-438/24-25 | |
| Other | Dichroic mirror | Semrock | FF458-Di02-25 × 36 | |
| Other | Emission filter | Semrock | FF01-483/32-25 | |
| Other | Emission filter | Semrock | FF01-542/27-25 | |
| Other | Filter wheel | Sutter Instruments | Lambda 10B | |

