## [Editor Report]

This manuscript shows that lactate is an important energy substrate and its metabolism shapes neuronal activity in the rodent somatosensory neocortex through K_ATP_ channels.

---

## [Decision Letter]

**Decision letter after peer review:**

Thank you for submitting your article "Lactate is a major energy substrate for cortical neurons and enhances their firing activity" for consideration by *eLife*. Your article has been reviewed by 2 peer reviewers, and the evaluation has been overseen by a Reviewing Editor and John Huguenard as the Senior Editor. The reviewers have opted to remain anonymous.

Essential revisions:

I request the Authors to revise the manuscript following all the suggestions raised by the Reviewers. I recommend to pay particular attention to the following points:

1) Improve the reporting to quantitatively assess the expression of Kir6.2 and SUR1 in various cortical cell types according to the experimental data.

2) Deemphasize the conclusions regarding co-expression of Kir6.2 and SUR1 subunits as well as the suggestion that neurons and beta cells use the same K_ATP_ channel, because both statements are supported only by a minority of data.

3) Improve the section related to K_ATP_ channel presence in neurons according to Reviewer 2 suggestions.

4) Provide a more complete rationale for the use of lactate concentration as detected in the blood and not in the CSF.

5) Discuss the discrepancy between limited (range: 19-28%) MCT1 and MCT2 expression and widespread lactate effects.

*Reviewer #1 (Recommendations for the authors):*

The authors could refer to the ribo-tag-based transcriptomics published by Doyle et al. (2008, Cell 10.1016/j.cell.2008.10.029) to support that many neuron cell types express Kir6.2 (kcnj11) and SUR1 (abcc8).

Obviously, in vivo verification of lactate-induced neuronal excitability via K_ATP_ would be ideal, but such a study is a project in itself, and it is not appropriate to request such a project as part of this revision.

As the K_ATP_ current largely depends on ATP/ADP concentrations, the discussion on fluctuations of these purines in the physiological condition, if known, may be good. However, this is a mere suggestion and is totally optional.

If there is a space to discuss limitations, perhaps the rat age being juvenile could be mentioned.

---

## [Author Response]

Essential revisions:I request the Authors to revise the manuscript following all the suggestions raised by the Reviewers. I recommend to pay particular attention to the following points:1) Improve the reporting to quantitatively assess the expression of Kir6.2 and SUR1 in various cortical cell types according to the experimental data.

We have improved the reporting on the qualitative detection for Kir6.2 (*Kcnj11*) and SUR1 (*Abcc8*) subunits in various cortical cell types (page 6, lines 26-32). To avoid confusion between detection and expression we now use only "detection" for scRT-PCR data and "expression" for functional data. Accordingly, in Figures 1F, 3B, 6A and 6—figure supplement 1, "Occurrence" was changed to "Detection rate".

2) Deemphasize the conclusions regarding co-expression of Kir6.2 and SUR1 subunits as well as the suggestion that neurons and beta cells use the same K_ATP_ channel, because both statements are supported only by a minority of data.

We have deemphasized the conclusion regarding co-expression of for Kir6.2 (*Kcnj11*) and SUR1 (*Abcc8*). We now provide the proportion of neurons in which both subunits were unambiguously co-detected (page 6, lines 30-32).

We also report the number of neurons pharmacologically characterized and analyzed by scRT-PCR (page 8, lines 8-12). All these neurons were found to express functional K_ATP_ channels though *Kcnj11* and *Abcc8* subunits were detected in only a minority of them. We thus conclude that scRT-PCR underdetected these mRNAs. Nevertheless, we also tone down the suggestion that neurons and beta cells utilise the same K_ATP_ channel.

3) Improve the section related to K_ATP_ channel presence in neurons according to Reviewer 2 suggestions.

We have improved the section related to K_ATP_ channel presence in neurons.

4) Provide a more complete rationale for the use of lactate concentration as detected in the blood and not in the CSF.

We agree with both reviewers' concerns. The rationale for the lactate concentration used was its isoenergetic condition to 10 mM glucose (page 10, lines 14-15). We also discuss the plasma vs CSF lactate concentration and the possibility that blood-borne lactate could participate in lactate-sensing (page 18, lines 9-19).

5) Discuss the discrepancy between limited (range: 19-28%) MCT1 and MCT2 expression and widespread lactate effects.

We have now discussed the discrepancy between MCT1 (*Slc16a1*) and MCT2 (*Slc16a7*) detection and the widespread lactate effects which are most likely due to their low abundance at the single cell level (pages 15,16, lines 32-34, 1-6). We also provide a counter example with *Ldh* subunits, expressed at higher single-cell levels, for which the higher scRT-PCR detection rate was found to match the functional data (page 16, lines 6-9).

Reviewer #1 (Recommendations for the authors):The authors could refer to the ribo-tag-based transcriptomics published by Doyle et al. (2008, Cell 10.1016/j.cell.2008.10.029) to support that many neuron cell types express Kir6.2 (kcnj11) and SUR1 (abcc8).

We thank the reviewer for this suggestion. Doyle et al. 2008 is now quoted in the discussion (page 13, lines 26-27).

Obviously, in vivo verification of lactate-induced neuronal excitability via K_ATP_ would be ideal, but such a study is a project in itself, and it is not appropriate to request such a project as part of this revision.

We fully agree that in vivo verification would be ideal and we are planning such a project in the near future.

As the K_ATP_ current largely depends on ATP/ADP concentrations, the discussion on fluctuations of these purines in the physiological condition, if known, may be good. However, this is a mere suggestion and is totally optional.

We now discuss the possibility that ATP/ADP ratio could fluctuate under physiological conditions (pages 16,17, lines 31-34, 1-7). However, and since enhanced firing was stable for several minutes, our data indicate that ATP levels were stable once firing was enhanced by lactate. A similar stable ATP level has been reported in pancreatic cells during high glucose stimulation (Tanaka et al. 2014, DOI: 10.1074/jbc.M113.499111). One potential mechanism is that increased energy consumption induced by spiking activity, is likely compensated by calcium-dependent energy metabolism. Indeed, intracellular calcium increase, notably associated with spiking activity, is known to stimulate mitochondrial ATP production through enhancement of the activities of several mitochondrial dehydrogenases of the Krebs cycle (reviewed in McCormack et al. 1990, DOI: 10.1152/physrev.1990.70.2.391). Nonetheless, when energy consumption is high, as during network synaptic transmission, fluctuations of ATP/ADP ratio and slow oscillation of spiking activity can occur, as reported in the entorhinal cortex (Cunningham et al., 2006, doi: 10.1073/pnas.0600604103).

If there is a space to discuss limitations, perhaps the rat age being juvenile could be mentioned.

We now briefly report that our observations have been made in juvenile animals and extend previous observation performed in neonates (page 13, line 2 and page 17, line 15). We also mention the necessity to validate the existence of lactate-sensing in the adult (page 17, lines 17-18).